# Effects of dopamine D2/3 and opioid receptor antagonism on the trade-off between model-based and model-free behaviour in healthy volunteers

**Nace Mikus[1,2]\*, Sebastian Korb[1,3], Claudia Massaccesi[4], Christian Gausterer[5], Irene Graf[6], Matthäus Willeit[6], Christoph Eisenegger[1‡], Claus Lamm[1], Giorgia Silani[4†], Christoph Mathys[2,7,8†]**

[1]Department of Cognition, Emotion, and Methods in Psychology, Faculty of Psychology, University of Vienna, Vienna, Austria; [2]Interacting Minds Centre, Aarhus University, Aarhus, Denmark; [3]Department of Psychology, University of Essex, Colchester, United Kingdom; [4]Department of Clinical and Health Psychology, Faculty of Psychology, University of Vienna, Vienna, Austria; [5]FDZ-Forensisches DNA Zentrallabor GmbH, Medical University of Vienna, Vienna, Austria; [6]Department of Psychiatry and Psychotherapy, Medical University of Vienna, Vienna, Austria; [7]Translational Neuromodeling Unit (TNU), Institute for Biomedical Engineering, University of Zurich and ETH Zurich, Zurich, Switzerland; [8]Scuola Internazionale Superiore di Studi Avanzati (SISSA), Trieste, Italy

**\*For correspondence:**
nace.mikus@univie.ac.at

[†]These authors contributed equally to this work

[‡]Deceased

[‡]Deceased on the 27th of February 2017.

**Abstract** Human behaviour requires flexible arbitration between actions we do out of habit and actions that are directed towards a specific goal. Drugs that target opioid and dopamine receptors are notorious for inducing maladaptive habitual drug consumption; yet, how the opioidergic and dopaminergic neurotransmitter systems contribute to the arbitration between habitual and goal-directed behaviour is poorly understood. By combining pharmacological challenges with a well-established decision-making task and a novel computational model, we show that the administration of the dopamine D2/3 receptor antagonist amisulpride led to an increase in goal-directed or 'model-based' relative to habitual or 'model-free' behaviour, whereas the non-selective opioid receptor antagonist naltrexone had no appreciable effect. The effect of amisulpride on model-based/model-free behaviour did not scale with drug serum levels in the blood. Furthermore, participants with higher amisulpride serum levels showed higher explorative behaviour. These findings highlight the distinct functional contributions of dopamine and opioid receptors to goal-directed and habitual behaviour and support the notion that even small doses of amisulpride promote flexible application of cognitive control.

## Editor's evaluation

This study provides novel evidence that a dopamine D2/D3 receptor antagonist enhances model-based control of behavior, whereas blocking opioid receptors has no effect. These conclusions are based on compelling behavioral and computational modeling data. The paper makes an important contribution to our understanding of how dopamine shifts the balance between two subsystems regulating behavior and may improve the understanding of motivational dysfunctions in mental disorders like addiction.

## Introduction

Several theories of decision making postulate the existence of two distinct systems that drive our behaviour: a habitual system, which is automatic, reflexive and fast; and a goal-directed system, which is deliberative, reflective and effortful (*Dickinson, 1985*; *Dolan and Dayan, 2013*; *Balleine and O'Doherty, 2010*). This dichotomy of systems has a computational analogue in 'model-free' and 'model-based' decision-making models (*Dolan and Dayan, 2013*; *Daw et al., 2005*). A model-free agent simply selects actions that have led to rewarding outcomes in the past. This strategy is fast, computationally cheap, but can be inaccurate. A model-based agent uses an internal model of the environment to flexibly plan behavioural responses. This leads to more goal-oriented behaviour but is slower and relies on effortful cognitive control.

Studies investigating how individuals allocate control between the two systems have shown that model-based control is increased when there is more to gain (*Patzelt et al., 2019*; *Kool et al., 2017*), and decreased when cognitive resources are scarce (*Otto et al., 2013b*; *Otto et al., 2013a*). Everyday behaviour can therefore be thought of as constant weighing of costs and benefits of applying model-based over model-free decision-making strategies (*Kool et al., 2017*; *Daw et al., 2011*; *Kool and Botvinick, 2018*). Failure to exert cognitive control over habitual urges in order to avoid negative outcomes has been suggested to be a hallmark of substance addiction (*Dolan and Dayan, 2013*; *Everitt et al., 2008*). In support of this, studies show that decreased model-based control is linked to stimulant use disorder (*Voon et al., 2015*) and seems to constitute a transdiagnostic dimensional trait related to compulsive behaviour across clinical (*Voon et al., 2015*; *Voon et al., 2017*) and non-clinical populations (*Gillan et al., 2016*). In light of increasing deaths from drug overdoses (*EMCDDA, 2020*; *Seth et al., 2018*), it is important to understand how different neurotransmitter systems affect the competition for cognitive resources when deciding between habitual and goal-directed actions.

Opiates, psychostimulants, and most other drugs of abuse increase the release of dopamine along the mesolimbic pathway (*Koob and Bloom, 1988*; *Di Chiara, 1999*), a circuit that plays a central role in reinforcement learning (*Schultz et al., 1997*). On top of this, the reinforcing properties of addictive drugs also depend on their ability to activate the μ opioid receptors (*Le Merrer et al., 2009*; *Benjamin et al., 1993*; *Becker et al., 2002*). This suggests that both the dopamine and the opioid systems might be particularly relevant in model-free reinforcement learning processes that drive the formation of habitual behaviour. Studies in rodents show that activating receptors of both systems across the striatum increases cue-triggered wanting of rewards (*Peciña and Berridge, 2013*; *Soares-Cunha et al., 2016*). Conversely, inhibition of both D1-type and D2-type dopamine receptors (referred to as D1 and D2 from here on) as well as opioid receptors reduces motivation to obtain or consume rewards (*Soares-Cunha et al., 2016*; *Laurent et al., 2012*; *Peciña, 2008*). This data raises the hypothesis that the drift towards habitual control is enabled by dopamine and opioid receptors via a common neural pathway.

Recent work in humans provides some evidence in this direction, whereby systemic administration of opioid and D2 dopamine receptor antagonists causes a comparable reduction of cue responsivity and reward impulsivity (*Weber et al., 2016*) and decreases the effort to obtain immediate primary rewards (*Korb et al., 2020*). This suggests that when allocating control between the model-based and model-free system, dopamine or opioid receptor antagonists might comparatively disrupt model-free behavioural strategies and increase model-based behaviour. Yet, no study in humans has directly investigated this. Furthermore, disrupting habit formation might not in itself lead to increased model-based control, without either increasing the perceived value of applying cognitive control or decreasing the cost of doing so.

Crucially, there are important differences in how each of the two neurochemical systems relate to the types of cognitive control that are pivotal for model-based behaviour. Across a wide range of studies using various dosing schemes, opioid receptor antagonists did not have an effect on tasks in which exertion of cognitive control plays a key role, such as working memory (*File and Silverstone, 1981*; *Volavka et al., 1979*; *Martín del Campo et al., 1992*), sustained attention (*Zacny et al., 1994*), or mathematical problem-solving (*Martín del Campo et al., 1992*) (see *van Steenbergen et al., 2019* for a review). Dopaminergic circuits, on the other hand, are central in a variety of higher cognitive functions and goal-directed behaviour (*Brozoski et al., 1979*). In particular, D1 dopamine receptors in the prefrontal cortex enable the maintenance of goal-relevant information and working memory (*van Schouwenburg et al., 2010*; *Sawaguchi and Goldman-Rakic, 1991*; *Williams and Goldman-Rakic,*

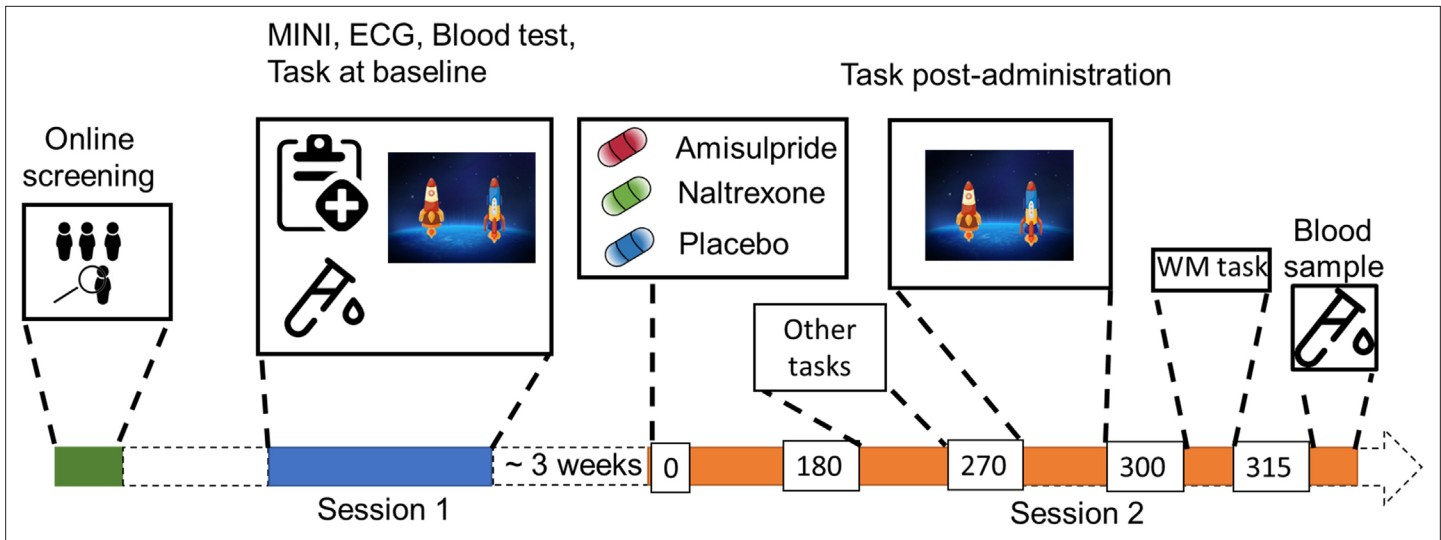

**Figure 1.** Study Procedure. After an initial online screening, participants were invited to the lab for a first visit (Session 1), where they were subjected to a medical check-up, before playing the two-step task for the first time. If they fulfilled the study criteria, they were invited for another visit (Session 2), where they received either 400 mg of amisulpride, 50 mg of naltrexone, or placebo (mannitol). After 180 min of waiting time, participants started with the test battery. Approximately 270 min after drug intake, participants performed the two-step task the second time, followed by a Reading Span (working memory) task, and a blood draw to determine amisulpride serum levels.

The online version of this article includes the following figure supplement(s) for figure 1:

**Figure supplement 1.** Side effects.

*1995*; *Goldman-Rakic, 1997*), while the D2 dopamine receptor activity disrupts prefrontal representations (*Durstewitz and Seamans, 2008*). In support of this, decreased working memory performance was observed after blocking prefrontal D1, but not prefrontal D2 receptors (*Sawaguchi and Goldman-Rakic, 1991*; *Arnsten, 2011*; *Seamans and Yang, 2004*). In humans, systemic administration of D2 antagonism increased the ability to maintain and manipulate working memory representations (*Dodds et al., 2009*; *Frank and O'Reilly, 2006*) and increased the value of applying cognitive effort (*Westbrook et al., 2020*). This data suggests that blocking D2 receptors, in contrast to blocking opioid receptors, could further facilitate model-based behaviour by enabling or encouraging flexible use of cognitive control.

This study compared the effects of the highly selective D2/3 dopamine receptor antagonist amisulpride with the opioid receptor antagonist naltrexone on the model-based/model-free trade-off in healthy volunteers. We tested 112 participants with a deterministic version of the two-step task (*Daw et al., 2005*; *Kool et al., 2016*), at baseline and after administering amisulpride (N=38, 400 mg), naltrexone (N=39, 50 mg), or placebo (N=35) in a randomized, double-blind, between-subject design (*Figure 1*). Based on the assumption that the dopamine and opioid systems support the allocation of control between habitual and goal-directed systems through overlapping neural circuits, we expected both antagonists to comparatively increase model-based relative to model-free behaviour. Furthermore, based on the hypothesis that blocking D2 receptors affects the cost/benefit analysis of applying the model-based strategy, we expected a stronger shift towards model-based behaviour following amisulpride administration.

The two-step task used to assess the model-based/model-free trade-off is a well-established incentivized paradigm, where participants need to make choices in the first stage, which influence their outcomes in the second stage. Each trial of the task starts in one of two possible first-stage states, each featuring a pair of spaceships (*Figure 2a*). Upon choosing a spaceship, participants were taken to one of two planets in the second stage, where they encountered an alien that gave them points, which were converted to money at the end of the experiment. In each pair of spaceships, one spaceship flew to the red planet and one to the green. This deterministic mapping from spaceships to planets, as well as which spaceships were paired together, stayed constant throughout the task. However, the points received on each planet changed independently with a Gaussian random walk. When playing the game, participants could therefore simply repeat the

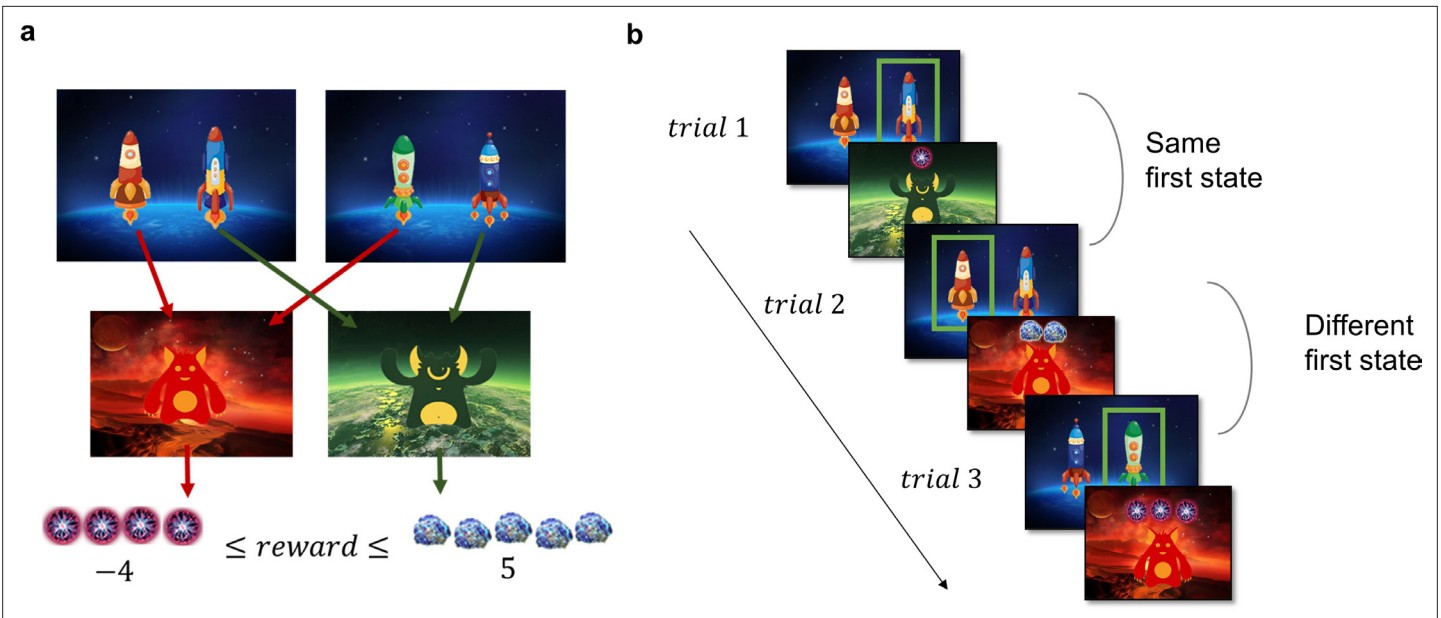

**Figure 2.** Task Design and Hypotheses. (**a**) In the first stage of the trial, the participants were presented with one of the two pairs of spaceships. Each of the spaceships flew to one of two planets in the second stage, where they encountered an alien that gave them points. The transition from each of the four spaceships to the planets was deterministic and stayed the same throughout the experiment. The points each alien gave changed independently according to a discretized Gaussian random walk with bouncing boundaries at –4 and 5. (**b**) Behaviour in the task showcasing trials where the previous first-stage state (spaceship pair) was the same (trial 2) and trials where it was different (trial 3). High number of points should encourage choosing the spaceship that flies to the same planet, what we term as 'staying with the previous choice'. Note that in the featured example, the participant after receiving –1 point in the trial 1 opted against staying with the previous choice in the next trial, and after receiving 2 points in trial 2 opted for staying with the previous choice, by choosing the spaceship, that flew to the same planet.

choice of spaceship that previously led to positive outcomes, regardless of which planet it was associated with (model-free behaviour, *Figure 2b*). Or they could attempt to remember which spaceship flew to which planet and choose the spaceship based on its associated planet (model-based behaviour, *Figure 2b*). The task tries to mimic real-life decision-making dilemmas, where we can use our knowledge about the world (e.g., a potentially dangerous virus is circulating) to override a habitual action (e.g., shaking hands when greeting someone) with a more appropriate one (e.g., greeting with an elbow bump or a bow).

As an initial straightforward approach to dissociate model-based and model-free behaviour, we first examined how previous reward points affected the probability to stay with the previous choice. We then turned to computational modelling and explicitly modelled the relative weight, $\omega$, of model-based compared to model-free behaviour and disentangled it from other processes that might affect participants' choices, such as exploration and reward devaluation. To assess any effects of the two pharmacological compounds on cognitive control, we also collected data on working memory performance through the Reading Span task (*Klaus and Schriefers, 2016*). We also aimed to control for changes in mood through a mood questionnaire delivered at drug intake and 3 hr later. Whereas the dose we chose for naltrexone provides sufficient occupancy of opioid receptors, the dose for amisulpride is limited by the potentially detrimental side-effects of D2 antagonists (*Takano et al., 2006*). To explore the variance of drug effects across participants, we examined amisulpride serum levels in the blood. Finally, previous research implicated several genotypes related to baseline dopamine function in model-based/model-free behaviour. To control for these baseline differences and to explore potential genotype drug interactions, we collected data on four genetic markers related to prefrontal and striatal dopamine levels. The data and the analysis scripts are available open-access at https://github.com/nacemikus/mbmf-da-op, (copy archived at swh:1:rev:4822b12aa33d8e5eb60d8ad5a-f2a0d3392e00e20; *Mikus, 2022*).

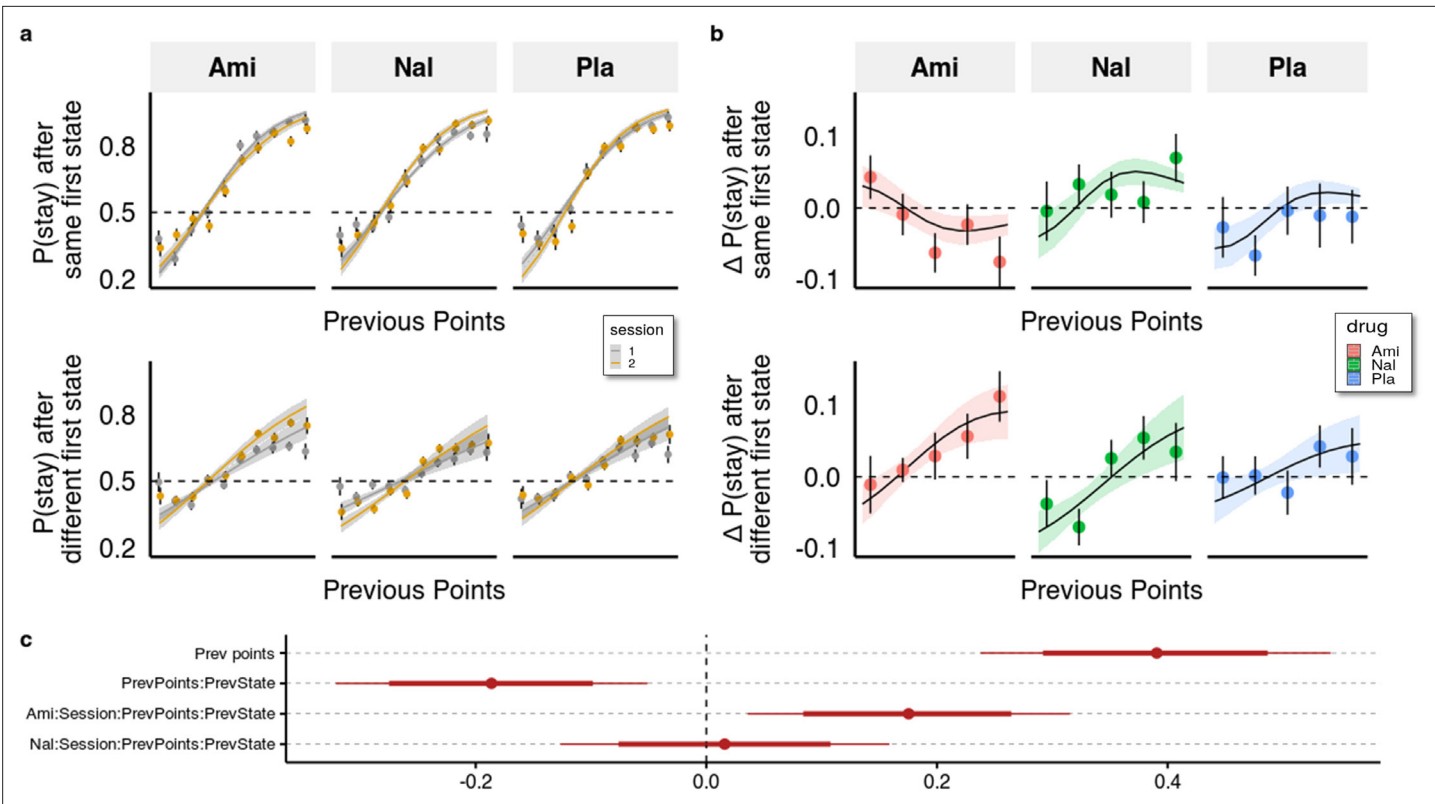

**Figure 3.** Behavioural analysis. (**a**) We used a hierarchical Bayesian logistic regression model to analyse how the probability to stay with the previous choice depended on previous points, previous first-stage state, session, and drug administration and their interaction. We allowed the intercept and the slopes for previous first-stage state and session to vary by participant (full table of coefficients in *Supplementary file 1a*). Points and error bars depict mean proportions and standard errors in trials where the participants stayed with their previous choice for each previous point averaged within session and drug group. Lines and ribbons depict the mean estimates and 80% credible intervals. When encountering a trial with the same first-stage state as in its preceding trial, participants were 1.495 times (95% CI [1.286, 1.739], P($\beta_{logodds} < 0$)<10e-3) more likely to stay with their previous choice for each additional previous point ($\beta_{logodds} = 0.402$, 95% CI [0.251, 0.553], P($\beta_{logodds} < 0$)<10e-3). In contrast, in trials where the first-stage state was different from the preceding trial, the odds (on the logarithmic scale) of repeating the previous choice of spaceship were reduced, with each additional point earned in the previous trial, by –0.204 (95% CI [–0.263, –0.146], P($\beta_{logodds} > 0$)<10e-3), and participants were only 1.219 times (95% CI [1.053, 1.410], P($\beta_{logodds}$ < 0)<10e-3) more likely to stay with their choice for each additional previous point. This indicates that participants often failed to consider the mapping from spaceship to planets when making choices in trials where the first-stage state differed from the previous trial. The effects of the drugs can be seen by the different slopes of previous points in the two sessions depicted for both trial types. (**b**) Differences in staying behaviour between sessions, binned into five different reward levels for clarity. Means with standard errors overlayed with means and 80% CI of estimated posterior distributions. (**c**) Means with 80% and 95% CIs of effect sizes (in logodds space) of selected regression coefficients of the hierarchical logistic regression model predicting staying behaviour from previous points (PrevPoints), previous first-stage states (PrevState), session, drug administrations, and all the interactions between them. Ami, amisulpride, n = 38; Nal, naltrexone, n = 39; Pla, placebo, n = 35.

## Results

### Effects of dopaminergic and opioidergic antagonism on staying with previous choices

One way to quantify the trade-off between the two systems is by looking at the interaction effect of previous points on staying behaviour in trials where knowledge about the task structure is irrelevant (same first stage state as in the previous trial) with trials where it is relevant (different first stage states as in the previous trial). With this, we control for more general effects of the drugs on decision-making behaviour and isolate the effect on the allocation of control between the two systems. As evident from *Figure 3* (and *Supplementary file 1a*), previous points increased the likelihood to stay with the previous choice ($\beta_{logods} = 0.391$), with a 95% credible interval (CI) [0.238, 0.542], and the proportion of the interval below zero P($\beta_{logodds}$ <0)<10e-3), however, this was significantly less the case in trials with different compared to the same first stage states ($\beta_{logodds} = -0.186$ (95% CI [-0.321,–0.051], P($\beta_{logodds}$

>0)<0.005). This indicates that participants often failed to consider the mapping from spaceship to planets when making choices in trials where the first stage state differed from the previous trial.

Compared to placebo, amisulpride significantly increased the difference between the effects of previous points on staying behaviour in different vs same first state trials, as indicated by a significant four-way interaction ($\beta_{logodds}$ = 0.176, 95% CI [0.036, 0.351], P($\beta_{logodds}$ <0)<0.007, *Figure 3*). When looking at each type of trial separately, we found that amisulpride also decreased the effect of previous reward points on the probability to stay when the first-stage state was the same ($\beta_{logodds}$=−0.140, 95% [CI=−0.274,,−0.009], P($\beta_{logodds}$ >0=0.02)), and slightly (and non-significantly) increased it when the first-stage state was different ($b_{logodds}$ = 0.036 (95% CI –0.091, 0.158), P($\beta_{logodds}$ <0)=0.285). For naltrexone, the log odds estimate of the four-way interaction coefficient was centred around 0.016, (95% CI [–0.127, 0.159], P($\beta_{logodds}$ <0)=0.41), indicating no marked difference between trial types; there were also no significant differences when looking at each trial-type separately (all p>0.32).

These results suggest that naltrexone did not affect the trade-off between goal-directed and habitual behaviour in this task. Amisulpride, on the other hand, increased performance in trials where keeping track of the mapping between spaceships and planets was advantageous, compared to those where it was not. However, somewhat counterintuitively, amisulpride also had a seemingly detrimental effect on decision-making overall, as indicated by the reduced effect of previous points on staying behaviour in trials with the same first-stage state. This pattern of behaviour might be due amisulpride's effect on both model-based/model-free behaviour and other decision-making processes, such as the tendency to explore other options. We note also that this analysis approach ignores one important aspect of the task, namely that subjects were comparing the relative points between two independently changing planets, thereby the points received in each trial should be seen in relation to the points that the participants expected to get were they to choose the other spaceship. To address these issues, we employed computational modelling.

## Estimation of drug effects with computational modelling

We defined a model (M1) where, similarly to the computational models previously used with this task (*Kool et al., 2017*; *Kool et al., 2016*), the trade-off between the goal-directed and habitual components is captured by the weighting parameter $\omega$, which embodies the degree to which the choice of participants on each trial is influenced by model-based ($\omega = 1$), or model-free ($\omega = 0$) subjective values of each of the spaceships. In this model, the subjective values of both the model-free and the model-based model components are defined as the last observed outcome following the choice of that spaceship. The crucial difference between the two components is that, whereas the model-free agent learns only by experiencing direct outcomes of spaceship selection, the model-based component always considers the deterministic mapping from spaceships to planets, and thus learns the subjective values of planets. We also include an inverse temperature parameter $\eta$ (lower $\eta$ indicates more explorative or noisy behaviour) and a *discounting parameter* $\gamma$ that marks the degree of devaluation of non-chosen (and non-encountered) spaceships for the model-free component in each trial.

We compared the above-described model to two Dual-systems Reinforcement Learning (RL) models (Fig. S1a). In these two models, the degree to which an outcome in each trial affects the subjective values of actions at each stage is determined by a learning rate parameter. The model-free agent thus learns the subjective action values in each stage from experience, by increasing the values of actions and states that lead to outcomes that were better than expected and decreasing the values when the outcome was worse than expected. We allow the learning rate at both stages to either be different (model M2) or the same (model M3). Note, that the model M1 is version of these RL model, where the learning rates are set to 1 and a devaluation parameter is included on the non-chosen option. This is motivated by the observation that the rewards change according to a Gaussian random walk (and are not probabilistic), and therefore the last encountered outcome is the best guess the agent can make. When comparing the performance of the three models we found that the model M1 has better out of sample predictive accuracy compared to the other two models (Fig. S1b). We verified the winning model with parameter recovery (Fig. S1c) and posterior predictive checks (Fig. S1d–f).

We first observed that the more model-based choices the participants made, the more money they earned (r=0.65, 95% CI [0.53, 0.76]). This serves as a validity check of the task, which was designed to make cognitive control pay off (literally) (*Kool et al., 2016*). We then looked at how the model parameters relate to the random slopes from the behavioural analysis of staying behaviour and found

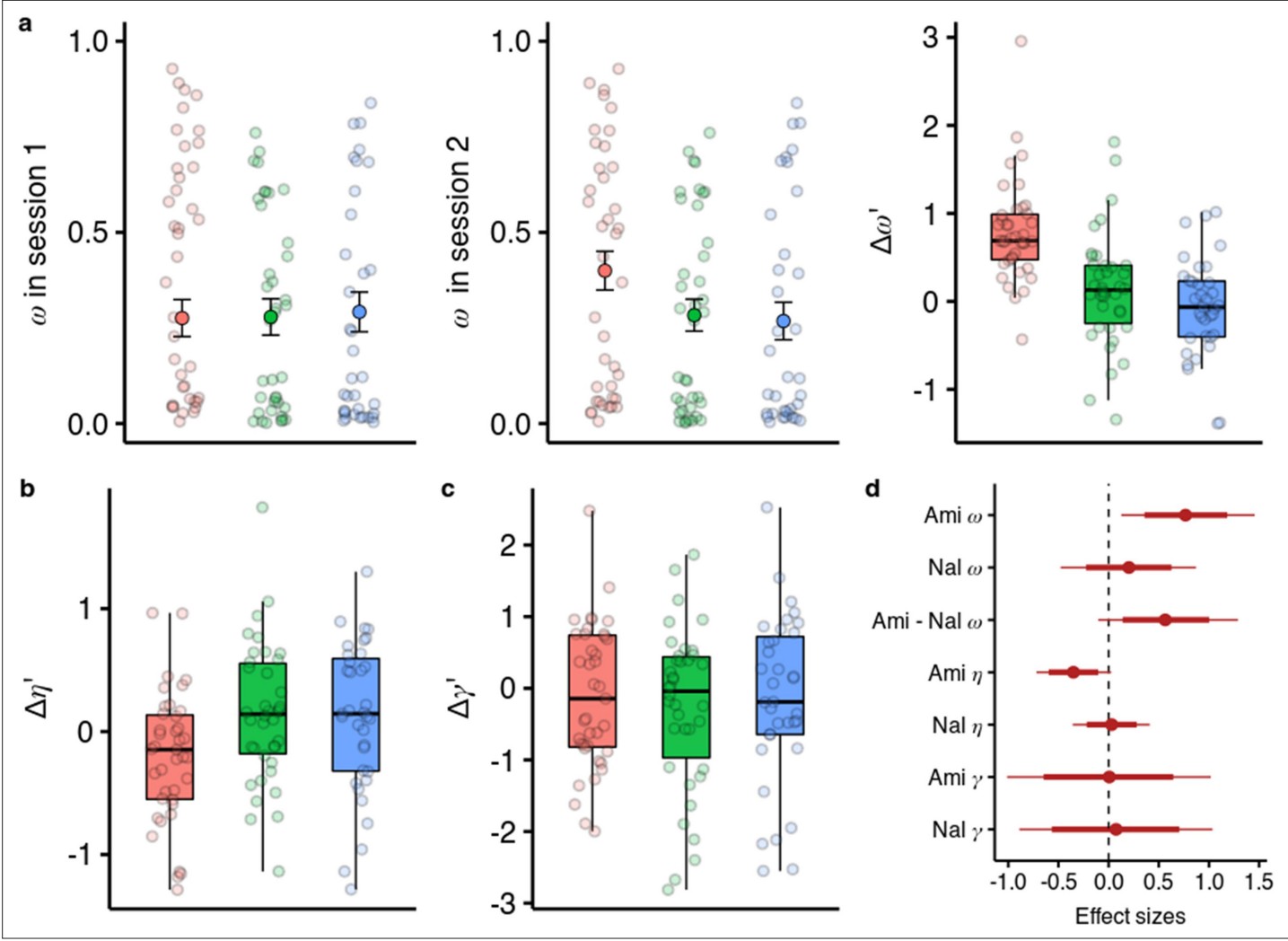

**Figure 4.** Effects of amisulpride and naltrexone on the model parameters. The best performing model (**M1**) is described with three free parameters, $\omega$ (the degree of model-based vs model-free value contributions to choice), $\gamma$ (the degree of devaluation of unencountered spaceships), and $\eta$ (the inverse temperature in the softmax mapping from values to probabilities). The parameters for both sessions and effects of drug treatments were estimated in one hierarchical model. (**a**) Going from session 1 to session 2, amisulpride administration led to higher estimations of $\omega$, and therefore increased model-based relative to model-free control. The difference between sessions of the model-based weights is shown in parameter estimation space (hence the prime). (**b**) Difference in the parameter $\gamma$. (**c**) Difference in the inverse temperature parameter $\eta$. Lower values mean higher exploration. (**d**) Posterior distributions of the effect of drugs, compared to placebo, on group-level mean session differences. Effect sizes with 80% and 95% CI. Ami, amisulpride, n = 38; Nal, naltrexone, n = 39; Pla, placebo, n = 35.

The online version of this article includes the following figure supplement(s) for figure 4:

**Figure supplement 1.** Computational modelling.

**Figure supplement 2.** Results of the model including stickiness parameters.

that the participant-level (random effect) slope for the effect of previous points on staying behaviour in different vs same first state trials was most strongly related to $\omega$ (d=0.493, p<10e-3) and negatively related to the inverse temperature parameter $\eta$ (d=–0.328, p<10e-3), and the slope for trials with same first states was mostly related to $\eta$ (d=0.822, p<10e-3), and less so to $\omega$ (d=0.235, p<10e-3).

## Dopaminergic antagonism increases model-based relative to model free control

We embedded the model parameter estimation within a hierarchical Bayesian inference framework and estimated the drug effects on all three parameters in one model (*Zhang et al., 2020*). We found

that under amisulpride the difference in $\omega$ between the two sessions is higher than in the placebo group ($\beta^{\Delta\omega}_{ami}$ = 0.787, 95% CI [0.131, 1.510], P($\beta^{\Delta\omega}_{ami}$ <0)=0.010, **Figure 4a**), with effect size d=0.758, (95% CI [0.126, 1.455]). In contrast, there was no session difference in $\omega$ between naltrexone and placebo ($\beta^{\omega}_{nal}$ = 0.238, 95% CI [–0.443, 0.856], P($\beta^{\omega}_{nal}$ <0)=0.250), with a marginally significant difference with a moderate effect size between the effects of the two compounds ($\beta^{\Delta\omega}_{ami-nal}$ = 0.578, 95% CI [–0.108, 1.337], P($\beta^{\Delta\omega}_{ami-nal}$ <0)=0.046, d=0.557, 95% CI [–0.104, 1.289]).

## Dopaminergic antagonism increases the exploration parameter in participants with high serum levels

We found no evidence that either the blockade of dopamine D2 or opioid receptors influenced the devaluation parameter $\gamma$ (**Figure 4c and d**, $\beta^{\gamma}_{ami}$=0.01, 95% CI [–1.00, 1.00], $\beta^{\gamma}_{nal}$=0.07, 95% CI [–0.88, 1.02]). However, we found some evidence for amisulpride effects on the inverse temperature parameter $\eta$ (**Figure 4b and d**, $\beta^{\eta}_{ami}$ = –0.33, 95% CI [–0.67, 0.02], P($\beta^{\eta}_{ami}$ >0)=0.034, d=–0.38, 95% CI [–0.78, 0.03]). This would imply that amisulpride increases 'explorative' choices or choices that the model would not predict based on estimated action values. To verify if our model performs worse for the amisulpride group we looked at how the pharmacological treatment predicts out-of-sample prediction accuracy and found no differences between groups (**Figure 4—figure supplement 1 and f**, **Supplementary file 1b**). We also note that the differences between sessions in $\omega$ and $\eta$ are not negatively correlated (r=0.12, 95% CI [–0.06, 0.30]), suggesting that the two effects are not related to each other.

We next looked at whether the effects of amisulpride were different based on the effective dose. We reran the parameter estimation including a variable for amisulpride blood serum levels as a group-level covariate in the hierarchical model (**Figure 5**). We used a categorical variable due to the highly skewed distribution of serum levels (**DeCoster et al., 2011**) (see Methods for details). We found that amisulpride increased $\omega$ in both the low serum group (b=0.919, 95% CI [0.216, 1.722], P(b<0)=0.006, d=1.013, 95% CI [0.238, 1.897]) and the high serum group (b=0.872, 95% CI [0.008, 1.853], P(b<0)=0.024, d=0.961, 95% CI [0.009, 2.041]), with no difference between the effects (P(b>0)=0.458). However, when looking at the effects of amisulpride on $\eta$ we found that it was not reduced in the low serum group (b=–0.105, 95% CI [–0.523, 0.348], P(b>0)=0.323, d=–0.096, 95% CI [–0.477, 0.317]), but was reduced in the high serum group (b=–0.492, 95% CI [–0.96,–0.033], P(b>0)=0.018, d=–0.45, 95% CI [–0.877,–0.03]), with a (non-significant) moderate effect size difference between the effects (b=–0.393, 95% CI [–0.918, 0.118], P(b>0)=0.066, d=–0.359, 95% CI [–0.838, 0.108]).

We also reran the behavioural analysis, predicting the likelihood to stay with the previous choice as before, but including the serum variable. In line with the results from the computational model, we found that amisulpride significantly increased the difference between the effects of previous points on staying behaviour in different vs same first state trials both in the low serum group ($\beta_{logodds}$ = 0.235, 95% CI [0.047, 0.43], P($\beta_{logodds}$ <0)=0.006), as well as in the high serum group, although to a lesser extent ($\beta_{logodds}$ = 0.143, 95% CI [–0.03, 0.323], P($\beta_{logodds}$ <0)=0.054). Conversely, in the high serum group, amisulpride decreased the effect of previous points in the same first state trials (b=–0.172, 95% CI [-0.34,–0.002], P($\beta_{logodds}$ >0)=0.024), mirroring the effect of the computational model. In the low serum group, amisulpride also decreased the effect of previous points on staying behaviour in the first state trials ($\beta_{logodds}$ = –0.147, 95% CI [–0.34, 0.036], P($\beta_{logodds}$ >0)=0.060). Note that, although the 95% CI contained values above 0, this represents a slight inconsistency with the results from the computational model, where the exploration parameter $\eta$ was not reliably reduced in the low serum group.

Lower values of $\eta$ as well as lower effect of previous points on behaviour implies that participants behaviour more stochastically (or noisily) and that the computational model cannot explain that behaviour based on estimated values of the two spaceships. To see whether some of the increased variances can be explained by the tendency of participants to repeat actions or choices regardless of outcomes, we extended the M1 serum model to include a response stickiness ($\rho$) and stimulus stickiness ($\pi$) parameter. We found some evidence that amisulpride in participants with high serum levels increased switching between responses, as indicated by the reduced $\rho$ parameter (b=–0.466, 95% CI [–0.962, 0.043], P(b>0)=0.039). This effect was not present in the low serum group (b=–0.158, 95% CI [–0.618, 0.311], P(b>0)=0.251), with a non-significant difference between the effects (b=–0.294, 95% CI [–0.723, 0.13], P(b>0)=0.082). The inclusion of the two stickiness parameters did not markedly

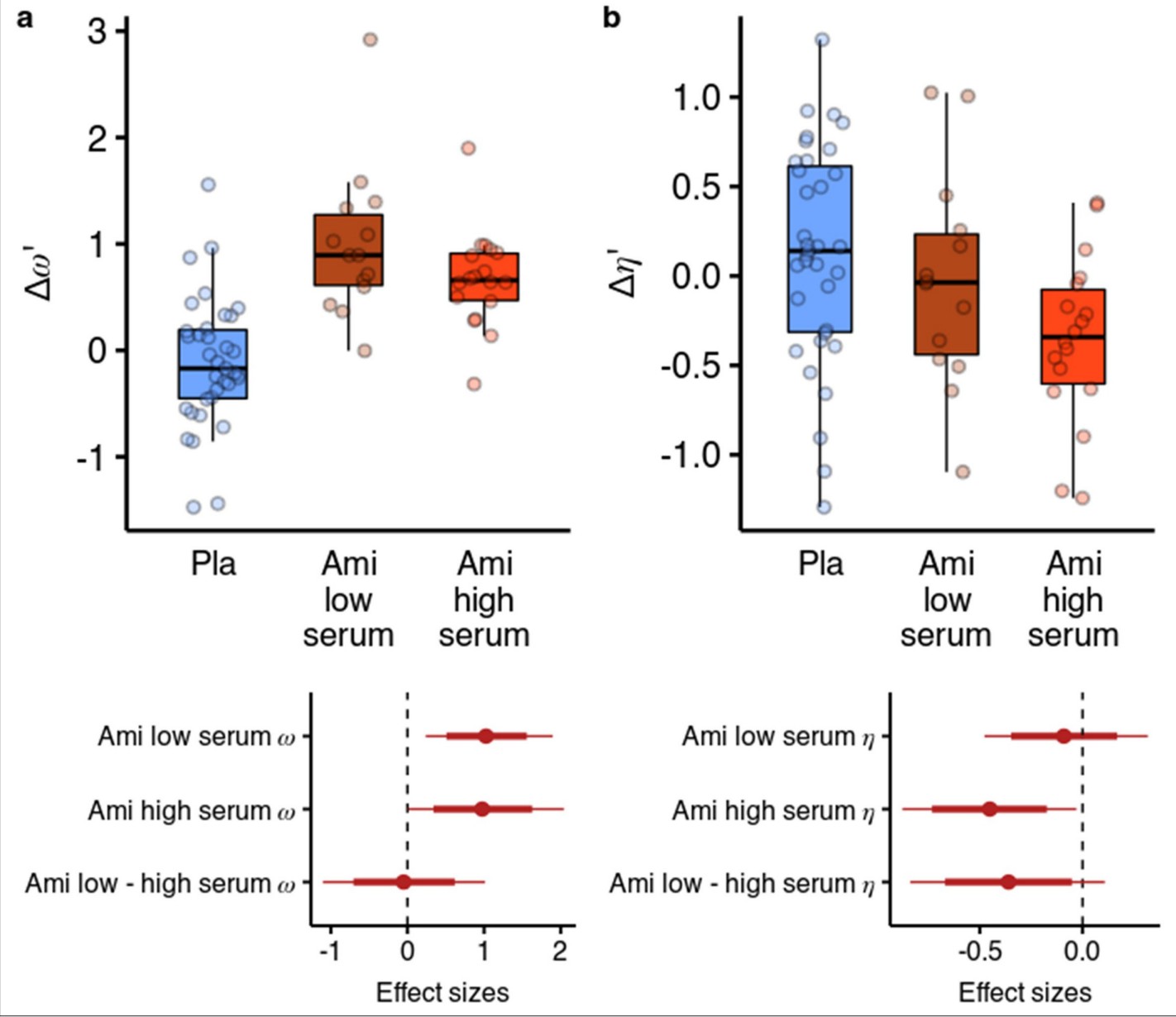

**Figure 5.** Effects of amisulpride across the high and low blood serum levels. (**a**) Across both effective dose groups of participants, amisulpride increased the model-free/model-based weight parameter. (**b**) Amisulpride increased exploration (decreased $\eta$) in the group that had high blood serum levels, but not in the group with low serum levels. Effect sizes with 80% and 95% CI. Ami, amisulpride; n = 32 (low serum, n = 14, high serum, n = 18); Nal, naltrexone, n = 39; Pla, placebo, n = 35.

change the posterior distributions of effects of amisulpride on $\eta$, but did reduce the effect on $\omega$ in the high serum group (b=0.571, 95% CI [–0.361, 1.527], P(b<0)=0.111, see *Figure 4—figure supplement 2* for other relevant posterior distributions). We note, however, that the model including stickiness parameters performed worse than the model without the stickiness parameters, with a lower out-of-sample trial-wise predictive performance ($\Delta elpd$ (sd) = –413 (31.5)).

## Working memory, mood, and genetics

To examine whether working memory capacity was affected by our drug manipulation, we analysed performance in the Reading Span task and found no evidence for drug effects on proportion of correct recalls ($\beta^{wm}_{ami}$ = –0.02, 95% CI [–0.07, 0.02], $\beta^{wm}_{nal}$ = –0.03, 95% CI [–0.08, 0.02]). We also found no difference in mood changes following amisulpride administration from drug intake and 3 hr in

either serum group (**Supplementary file 1cd and e**), but found that naltrexone flattened both positive (b=–0.411, 95% CI [–0.855, 0.029], P(b>0)=0.035) and negative affect (b=–0.524, 95% CI [-1.001,–0.061], P(b>0)=0.013). To explore what other group level variables might influence the drug effects, we ran two linear models predicting differences in $\omega$ and $\eta$ between sessions including working memory and mood ratings as covariates in the analysis and Sex, Weight and Age as moderator variables. We found no significant effects and neither did conditioning on the covariates affect inference of drug effects on $\omega$ or $\eta$ (**Supplementary file 1f and g**).

We also found no robust effects of genotype variables on either baseline measures or on the drug effects on either $\omega$ or $\eta$ (Supplementary Files h, i). In line with previous studies using similar paradigms, we used the *COMT* single nucleotide polymorphism (SNP) as a marker for prefrontal dopamine function, and *DARPP-32* SNP as a marker of striatal D1 receptor function (**Doll et al., 2016**; **Männistö and Kaakkola, 1999**; **Trantham-Davidson et al., 2004**). We also investigated two genetic markers for striatal dopamine levels, namely the *Taq1A* SNP and the *DAT1* polymorphism, a 40 base-pair variable number tandem repeat polymorphism (VNTR) of the dopamine transporter (**Laakso et al., 2005**; **Eisenegger et al., 2013**; **Eisenegger et al., 2014**) (see Supplementary Note 1 for details on genotypes and how they relate to striatal and prefrontal dopamine function).

## Discussion

Drugs that stimulate opioid or dopamine neurotransmitter systems can lead to compulsive drug-taking, which can result either from increased reliance on habits or from failures to exert cognitive control over habitual urges. Experimental studies strongly implicate both the dopamine and opioid neurotransmitter systems in processes related to habit formation. Here, we asked whether pharmacologically blocking opioid and dopamine receptors promotes the use of goal-directed over habitual control in healthy volunteers. Using the two-step task and a novel computational model, we found that blocking D2-like dopamine receptors increases the weight on model-based relative to model-free control, whereas blocking opioid receptors has no appreciable effect, with a marginally significant difference between the effects (with a moderate average effect size). Additionally, we found that amisulpride increases choice exploration, particularly in the group of participants with high amisulpride serum levels in the blood. The results from the computational model mirrored those obtained when analysing the effects of previous points on staying with prior choices. The degree to which reward in a previous trial affected choice repetition in trials with the same first stage states was mostly related to the exploitation/exploration parameter in the model and was reduced under amisulpride. Conversely, the relative increase in the effect of previous points on choice repetition between different vs same first stage state trials was more related to the model-based/model-free weight and was increased under amisulpride. Our findings provide initial evidence for a divergent involvement of the dopamine and opioid neurotransmitter systems in the shift between habitual and goal-directed behaviour. The lack of effects of naltrexone on the model-based/model-free trade-off also provides some support for the notion that simply disrupting neurobiological systems that subserve habitual behaviour might not be enough to increase goal-directed behaviour in this task. An increase in the model-based/model-free weight following amisulpride administration advocates for dopamine playing a decisive role in flexibly applying cognitive control to facilitate model-based behaviour and highlights the specific functional contribution of the D2 receptor subtype.

Prior research on dopamine's involvement in the arbitration between the two systems has produced inconsistent results. Elevated dopamine levels, either at baseline or following pharmacological treatments such as L-dopa, are related to increased model-based behaviour in some studies (**Wunderlich et al., 2012**; **Deserno et al., 2015**; **Sharp et al., 2016**), but show no correlation or even opposite effects in others (**Kroemer et al., 2019**; **Voon et al., 2020**). This suggests that the effect of dopamine promoting agents on model-based/model-free trade-off might depend on the relative contributions of the various region-specific dopamine receptor subtypes. Studies have repeatedly shown that prefrontal dopamine circuits enable goal stability and distractor avoidance, primarily through D1 dopamine receptors (**van Schouwenburg et al., 2010**; **Sawaguchi and Goldman-Rakic, 1991**; **Williams and Goldman-Rakic, 1995**). According to the dual-state theory of prefrontal dopamine function (**Durstewitz and Seamans, 2008**), a state dominated by D1 receptor activity is characterized by a high-energy barrier and supports stabilization of prefrontal representations. Conversely, a D2 receptor dominated state has a low-energy barrier and facilitates flexible switching between representational

states. In line with this, D2 antagonists impair behaviour in tasks that require constant attentional set shifting (*Mehta et al., 2004*), but might improve performance in tasks where prefrontal goal representations are required (*Kahnt et al., 2015*). For example, D2 antagonism improves the ability of participants in certain cognitive tasks that require manipulation of task-relevant information in the working memory (*Dodds et al., 2009*; *Frank and O'Reilly, 2006*), but does not improve working memory capacity, or memory retrieval (*Dodds et al., 2009*; *Naef et al., 2017*; *Mehta et al., 1999*). The effects of D2 antagonism on model-based/model-free behaviour in our study can be interpreted within this framework to result from increased ability to maintain prefrontal representation of the mapping between the spaceships and the planets online. However, this is difficult to reconcile with the fact that model-based behaviour in dynamic learning paradigms, such as the one used here, also requires flexible updating of action values.

Theories of adaptive control emphasize that the prefrontal regions subserving cognitive control interact with striatal dopaminergic circuits (*McNab and Klingberg, 2008*; *Frank, 2005*). In the striatum, phasic dopaminergic bursts subserve prediction error propagation and action initiation (*Frank et al., 2004*) mainly, but not exclusively, through D1 dopamine receptors (*Soares-Cunha et al., 2016*; *Frank et al., 2004*; *Montague et al., 1996*). Slower baseline dopamine changes in tonic activity are believed to invigorate physical action (*Niv et al., 2007*) based on cost-benefit tradeoffs (*Salamone et al., 2016*). More recent work extended this framework and proposed that the same circuits subserve cost-benefit analysis of applying cognitive effort, framing the role of striatal dopamine as mediating the decision to apply cognitive control based on perceived value vs costs of doing so (*Westbrook et al., 2021*; *Cools, 2016*; *Cools, 2019*). In a recent study that combined neurochemical imaging with pharmacological administration of 400 mg of sulpiride (another D2 dopamine receptor antagonist) and methylphenidate (a drug that boosts striatal dopamine availability) showed that both drugs comparatively increased the willingness to exert cognitive effort, particularly in subjects with low striatal dopamine availability (*Westbrook et al., 2020*). An important aspect of drugs that target D2 receptors is that they are known to mainly act on presynaptic D2 receptors at low doses, while binding to postsynaptic receptors prevailingly accounts for their effects at higher doses (*Schoemaker et al., 1997*). Presynaptic D2 receptors are believed to play an autoregulatory control of dopamine signalling through inhibition of synthesis and dopamine release (*Ford, 2014*). In our data, we find that the increase in model-based control did not scale with higher effective doses and was present in participants with either high or low serum levels in the blood. The effect in the high serum group was slightly reduced (and not robustly above zero, p=0.111) in the computational model that included stickiness parameters, although we note that including stickiness parameters led to a poorer model fit. Taken together, this provides some evidence for the notion that the effects of amisulpride on model-based behaviour are not driven by postsynaptic effects but might rather be due to amisulpride increasing dopamine levels through presynaptic D2 receptor blockade and thereby boosting the willingness to apply cognitive effort.

Interestingly, amisulpride also increased choice stochasticity parametrised by the softmax inverse temperature parameter. In a paradigm with two choice options, it cannot be definitively determined whether this indicates higher decision-noise or increased exploration of alternative choices. We can however speculate that increased decision noise would lead to overall detrimental effects on learning in both trial types with same and different consecutive first stage states, which we do not observe in our data. The effect on the choice stochasticity parameter was only present in participants with a higher effective dose (*Rosenzweig et al., 2002*), suggesting that the effect was more likely to be post-synaptic. Similarly, in the same effective dose group, we found some evidence that amisulpride reduces response stickiness indicating increased switching between actions. This is well in line with a prominent model of the cortico-striatal circuitry implicating post-synaptic D2 receptors in exploration/exploitation (*Frank, 2005*) and supported by empirical data. In animal studies, activation of D2 receptors was shown to lead to choice perseverance and more deterministic behaviour, whereas D2 receptor inhibition increases the probability of performing competing actions and increases randomness in action selection (*Sridharan et al., 2006*). In humans, a recent neurochemical imaging study showed that D2 receptor availability in the striatum correlated with choice uncertainty parameters across both reinforcement learning and active inference computational modelling frameworks (*Adams et al., 2020*). Increased choice uncertainty was also observed in a social and non-social learning tasks in a study using 800 mg of sulpiride, a dose that is known to exert post-synaptic effects (*Eisenegger*

*et al., 2014*; *Mikus et al., 2022*). We note, however, that the evidence for the difference in exploration between the low and high serum groups was not robust (p=0.066). Furthermore, it has been suggested that increased striatal dopamine is also related to tendency for stochastic, undirected exploration (*Gershman and Uchida, 2019*; *Frank et al., 2009*), arising due to overall uncertainty across available options (*Gershman and Uchida, 2019*) or through increasing the opportunity cost of choosing the wrong option (*Niv et al., 2007*; *Cools, 2016*). This suggests that the same biological signature that leads to increased cognitive effort expenditure also promotes choice exploration. In line with this, both prior studies that investigated the effect of increasing dopamine availability with L-DOPA on model-based/model-free behaviour observed increase choice exploration as well as increased model-based behaviour (although in one it was only present in individuals with a higher working memory capacity) (*Wunderlich et al., 2012*; *Kroemer et al., 2019*).

The lack of a pronounced effect of naltrexone on model-based/model-free behaviour is unlikely to be due to issues of dosage or timing, since 50 mg of naltrexone leads to a~90% of μ receptor occupancy even 48 hr post intake (*Lee et al., 1988*; *Trøstheim et al., 2022*), with μ receptors being the primary (but not only) opioid receptor type that the drug binds to. One possibility is that opioid receptors are not crucial for model-free learning required in this task. This is in opposition to previous studies showing that acute administration of naltrexone, comparably to amisulpride, causes a reduction of cue responsivity and reward impulsivity (*Weber et al., 2016*), decreases effort to obtain immediate primary rewards (*Korb et al., 2020*), and decreases the wanting of rewards (*Soutschek et al., 2021*). Similarly, preclinical studies in rats show that naltrexone and naloxone, another opioid antagonist, decreased sucrose reinforced place preference (*Delamater et al., 2000*; *Agmo et al., 1995*). Furthermore, both naltrexone and naloxone are used to reduce craving in patients with substance use disorder (*Quednow and Herdener, 2016*). However, there is also evidence that opioid receptors are important for goal-directed behaviour. In particular, a study in rats showed that opioid receptor blockade leads to decreased sensitivity to reward value and accelerated habitual control of actions (*Wassum et al., 2009*), and in a recent neurochemical imaging study in humans μ opioid receptor availability correlated with goal-directed behaviour in a loss-only version of the two-step task (*Voon et al., 2020*). In fact, opioid agonists (but not antagonists) can in some cases lead to increased performance in cognitive control tasks (*van Steenbergen et al., 2019*). One explanation of our results is that naltrexone simply reduced the intrinsic value of reward and therefore decreased the motivation to exert cognitive effort, which would be in line with the observed flattening of both positive and negative affect after naltrexone administration.

An important limitation of the experimental approach used here is that it rests on the assumption that there are only two ways of learning, which moreover can be distinguished clearly. This assumption has been questioned in the reinforcement learning literature (*Feher da Silva and Hare, 2020*; *Daw, 2018*), and mirrors the scepticism of the two-system division of decision making in cognitive psychology (*Melnikoff and Bargh, 2018*). Although we made sure that participants understood what the task rules were and how they could maximize their gains, we cannot exclude the possibility that participants searched for alternative models to obtain rewards, such as different pressing patterns or simply favouring one stimulus over the other (*Feher da Silva and Hare, 2020*). In fact, the increased exploration parameter in the amisulpride group could be due to participants' increased exploration of this model space. Importantly, it should also be acknowledged that the behavioural setup in our study does not allow us to draw definite conclusions about the mechanisms that mediate amisulpride's effects on model-based or model-free behaviour. For example, it is not clear whether amisulpride increases the perceived benefit of applying cognitive control, or whether it increases the participant's ability to do so through various possible complementary processes, such as goal maintenance or planning abilities. Future studies should further elucidate the mechanistic contributions of dopamine receptors to the distinct coding and utilisation of task relevant representations (*Langdon et al., 2018*; *Stalnaker et al., 2019*).

One of the strengths of our design is a baseline measure, and the fact that the participants were all introduced to the task under no administration, thus avoiding potential effects of the treatment on task training. Although this design allowed us to reduce between-subjects variability, we cannot completely exclude order effects. Although unlikely, it is possible that the effects of the treatment that we observe come indirectly from the effects of the two drugs on either skill transfer from the previous session, or simply on the effect of the drugs on the part of the experiment that preceded the

task. For instance, participants under amisulpride could be less tired from other tasks and therefore more willing to exert effort in the task presented here. Speaking against this is the observation that we found no differences in mood between amisulpride and placebo regardless of low or high serum levels.

We note that opioid and dopamine systems are not the only neurotransmitter systems that have been implicated in the model-based/model-free tread-off. Most notably, depletion of the serotonin precursor tryptophane biases behaviour towards habitual and away from goal-directed (*Worbe et al., 2015*), suggesting that serotonin supports goal-directed behaviour. This claim is also in line with a recent neurochemical imaging study that showed that serotonin terminal density in the striatum correlated with higher goal-directed behaviour in a version of the two-step task (*Voon et al., 2020*) and decreasing prefrontal serotonin levels has been shown to promote drug seeking behaviour (*Pelloux et al., 2012*). In light of this, we also note that amisulpride also blocks serotonin receptors, albeit with lower affinity, which should be kept in mind when considering the findings of our study. In conclusion, we provide a first comparison of the contributions of the dopamine D2 and opioid receptors to arbitration between the model-based and model-free systems. Our results suggest that D2 dopamine antagonists might promote goal-directed behaviour when alternative habitual choices are available, already in low doses, while opioid blockade does not have such an effect on model-based behaviour. These findings are a step forward in understanding how neuromodulators control the arbitration between habitual and goal-directed decision-making systems, which can in the long run be crucial for developing targeted pharmacological treatments for addiction and other disorders of compulsivity.

## Methods

### Procedure

The study took place in the Department of Psychiatry and Psychotherapy at the Medical University of Vienna. After initial online screening, the participants were invited for a first visit, where they underwent a physical examination (electrocardiogram, hemogram) and a psychiatric screening before being subjected to the two-step task that consisted of a 25 min automated training period followed by 200 trials of the task (see *Figure 1* for the study outline). Participants eligible for the study were invited back to the clinic approximately 3 weeks after their first visit.

In their second visit participants received either placebo, 400 mg of amisulpride, or 50 mg of naltrexone. After a waiting period of 3 hr, participants solved two additional tasks (explained elsewhere *Korb et al., 2020*) before performing the two-step task (on average 4 hr 36 min, SD = 22 min, after the pill intake) as well as a working memory task. Waiting times and the doses of drugs were chosen based on previous pharmacological studies with the same compounds (*Weber et al., 2016*). To ensure comparable absorption, participants were asked to come with an empty stomach and received a standardized meal before pill intake. Participants filled out a questionnaire assessing mood and side effects right after pill intake and 3 hr later. There were no profound differences in side effects across the three drug groups, apart from a trend level effect of amisulpride on tiredness (*Figure 1—figure supplement 1*). Blood plasma levels confirmed amisulpride levels above 212.6 µg/l (mean (sd)=548.8 (96.9)), in all participants of the amisulpride group. For 7 subjects serum levels were not possible to calculate due to technical issues with blood samples. For 18 participants the value of amisulpride was at maximum (>604 µg/l), leading to a highly skewed distribution, with no variance in the upper 50% quantile. Because of this, we dichotomized the data into a categorical variable, whereby the high serum group consisted of participants with serum levels above 604 µg/l (n=18) and the low serum group had levels between 212.6 µg/l and 602.5 µg/l (n=14).

### Participants

Data were collected from 120 volunteers and constituted a subset of volunteers involved in a larger study (*Korb et al., 2020*). All participants were assigned an initial screening code and a subject ID number. For six participants the task data from the first session were lost due to failures in assigning screening codes from the first session to subject ID numbers in the second. For two participants the task data from the second session were lost owing to technical issues, resulting in 112 participants included in the main analysis (see *Supplementary file 1l* for exact subject numbers for each analysis).

All participants had no history of drug abuse or other psychiatric disorders and were matched in age, sex, and BMI (*Supplementary file 1j*). The study was approved by the Ethical Committee of the Medical University of Vienna (EK N. 1393/2017) and was in line with the Declaration of Helsinki (*World Medical association Declaration of Helsinki, 2013*). Participants received a monetary compensation of 90€ plus the extra money earned in the task.

## Genotypic analysis

Peripheral blood was collected by lancet and stored on Whatman FTA micro cards (Sigma-Aldrich). DNA was extracted using the QIAamp DNA Mini kit (Qiagen, Hilden, Germany). The VNTR polymorphism in the *DAT1* gene was investigated by PCR with 5'-fluorescent-dye-labeled forward primer and automated detection of PCR products by capillary electrophoresis (details of the procedure provided in the supplement). The single base primer extension (SBE) method also known as SNaPshot minisequencing was applied for the typing of single nucleotide polymorphism (SNP) variants (details provided in supplement). Accordingly, five informative SNPs [*DRD2/ANKK1 Taq1A* (*rs1800497*), *BDNF* (*rs6265*), *CDH13* (*rs3784943*), *OPRM1* (*rs1799971*) and PPP1R1B/DARPP-32 (*rs907094*)] were analysed simultaneously applying a multiplex strategy for PCR and SNaPshot minisequencing of purified PCR products. Typing of Val158Met variants (*rs4680*) in the *COMT* gene was carried out separately, applying a singleplex approach for PCR and SNaPshot. The *OPRM1* and *BDNF* SNPs were not included in the analysis in this task. Genetic data from three participants were lost. This led to the group distributions depicted in *Supplementary file 1k*. For details on the analysis see Supplementary Note 2.

## Task design

In the task, participants made an initial choice at stage one which took them to one of the two possible states ('planets') in stage two. There were two possible states in the first stage, each featuring a pair of spaceships. Each of the spaceships flew deterministically to one of the two planets where they encountered an alien who gave them either positive or negative points. The points in each planet changed according to a Gaussian random walk in the interval [–4, 5], rounded to whole numbers. Importantly, since both first-stage states led to the same two possible second-stage states, participants could transfer knowledge from the pair of spaceships in one first-stage state to the pair in the other. According to the conventional definition adopted here, a completely model-free agent relies entirely on its direct experience and will choose a spaceship based only on its reward history in previous trials featuring the same first-stage state, regardless of their experience with the other pair of spaceships. A completely model-based agent however will use the causal structure of the task to update the value of the spaceships based on which planet they fly to. This means that knowing that spaceship A and C fly to the same planet (but appear in different first step states) enables the model-based agent to learn about spaceship C by experiencing the outcome of its choice of spaceship A.

Participants had 2000 ms to choose in the first step and then again 2000ms to press the space bar once they had encountered the alien. Each acquired point was translated to 4 Eurocents and added to the overall compensation of the participant at the end of the experiment. The story and a thorough training session were employed to increase the comprehension of the task, as done before (*Decker et al., 2016*). To avoid all participants behaving in a model-based way (*Feher da Silva and Hare, 2020*) we increased the difficulty of the task by dynamically changing the drift of the reward-determining Gaussian walks. The Gaussian random walks, therefore, had various drift rates (0.5, 1, and 2). We generated one trajectory and shuffled it around to create two different testing sessions of similar difficulty. Each participant first went through an automated rigorous explanation of the task followed by 25 practice trials before completing 200 trials of the task.

## Behavioural analysis

To regularize our inference, we used a hierarchical Bayesian approach in all our analyses. We report estimates in parameter estimation space and indicate the precision of our estimates with credible intervals (*Zhang et al., 2020*; *Kruschke, 2014*). We also report the proportion of the credible interval that is above or below zero. Model-agnostic analysis of behaviour focused on predicting the probability of staying with the previous choice (choose the spaceship that flies to the same planet as in the previous trial). Behaviour was analysed using the *brms* package in R (*Bürkner, 2017*) which employs the probabilistic programming language Stan (*Carpenter, 2017*). We fit a binomial model

**Table 1.** Prior distributions for the behavioural analysis.

| | |
|---|---|
| Standard deviations | $\sigma \sim Half\ Cauchy\ (0, 2)$ |
| Regression coefficients | $\beta \sim N\ (0, 3)$ |
| Intercept | $\beta_0 \sim Student\ (3, 0, 10)$ |
| Prior for the correlation matrix | $R \sim LKJcorr\ (2)$ |

that predicted staying with the previous choice from reward obtained in the previous trial, modulated by the previous state (same or different), drug treatment, and session. The effects of session, previous state, previous points and all interactions between them were drawn from a multivariate normal distribution (were considered as correlated random effects). We report 95% credible intervals of estimates on the log-odds scale. Parameters were estimated with 4 chains, every 3000 iterations (1000 warmup), with priors listed in *Table 1*.

The number of participants used in the analysis depends on whether both sessions were included and whether genetic data were used (see *Supplementary file 1I* for an overview). Analysis with amisulpride blood serum levels included a categorial variable for the serum. A categorical variable was chosen due to the skewed distribution of serum levels. To relate behaviour to the parameters of the computational model we extracted the random slopes for each participant for the second session for both the effects of previous points on staying behaviour in trials with the same first stage state as in the previous trial, as well as the random slopes for the different vs same first state trials. We then standardized all slopes and the computational parameters and used two separate linear models to predict both slopes from the three computational parameters.

## Computational models

We defined three computational models that define the subjective values $Q$ of participants' action $a$, in trial $t$, with a first stage state $s$, as the weighted average of model-based ($Q_{MB}$) and model-free ($Q_{MF}$) subjective values.

$$Q\ (a, s, t) = \omega Q_{MB}\ (a, s, t) + (1 - \omega)\ Q_{MF}\ (a, s, t)$$

where $\omega$ is the weighting parameter (larger influence on choice of model-based values is indicated by $\omega$ being close to 1 and model-free control by $\omega$ being close to 0). The subjective values are then mapped on to probability of choosing $a$, and not $a'$, with the soft-max transformation:

$$P(a,\ s,\ t) = \frac{e^{\eta * Q(a,s,t)}}{e^{\eta * Q(a,s,t)} + e^{\eta * Q(a',s,t)}}$$

where $\eta$ is the *inverse temperature* parameter, that determines the stochasticity of choices and the exploration-exploitation trade-off. In the models that include stickiness parameters, the value function for action $a$ was extended as follows:

$$Q\ (a, s, t) = \omega Q_{MB}\ (a, s, t) + (1 - \omega)\ Q_{MF}\ (a, s, t) + \rho * resp\ (a) + \pi * stim\ (a)$$

where the indicator functions were defined as being 1 for the action that was the same as in the previous trial (*resp* $(a)$) or the stimulus that was chosen in the previous trial (*stim* $(a)$). The $\rho$ and $\pi$ parameters, therefore, determine the degree to which previous actions (or stimuli) tended to be repeated (*Kool et al., 2016*).

## Deterministic learning model (M1)

Because the outcomes at the second stage are determined by Gaussian random walks, the objectively best prediction by the agent is the last encountered outcome. We defined a model (M1), where the model-free subjective value of each of the spaceships is the last encountered outcome following the choice of that spaceship, and the model-based subjective value of each spaceship is the subjective value of the last encountered outcome following the planet that this spaceship flies to. This means

that the update equation of the model-free agent after receiving outcome $r(s_2, t)$ following an action $a$, in trial $t$, with first state $s_1$, is defined as

$$Q_{MF}(a, s_1, t) = r(s_2, t).$$

In the same trial, the value of the for the unchosen actions across both first level states are shrunk towards 0 with forgetting parameter $\gamma \in [0, 1]$:

$$Q_{MF}(a', s_1, t) = (1 - \gamma) \, Q_{MF}(a', s_1, t - 1),$$

$$Q_{MF}(a, s_1', t) = (1 - \gamma) \, Q_{MF}(a, s_1', t - 1),$$

$$Q_{MF}(a', s_1', t) = (1 - \gamma) \, Q_{MF}(a', s_1', t - 1),$$

where $a'$ is the unchosen action and $s_1'$ is the unencountered state. The model-based agent, on the other hand, after receiving outcome $r(t)$ following an action $a$, in trial $t$, with first state $s_1$, updates not only the experienced first stage state but also the unencountered first stage which would have led to the same outcome:

$$Q_{MB}(a, s_1, t) = Q_2(s_2, t) = r(s_2, t) \ ,$$

$$Q_{MB}(a, s_1', t) = Q_2(s_2, t) = r(s_2, t) \ ,$$

where, $Q_2(s_2, t)$ is the subjective value of the second stage state that action $a$ deterministically leads to.

## Dual-system reinforcement learning models (M2 and M3)

We compared our model M1 to two versions of a dual-system reinforcement learning model inspired by *Kool et al., 2017*; *Kool et al., 2016*. In these models, the model-free agent learns the subjective values of spaceships and planets through a temporal difference-learning algorithm (*Daw et al., 2011*; *Sutton and Barto, 1998*). The model-free agent was defined by 3 free parameters: the learning rate at the first stage ($\alpha_1$), and the second stage ($\alpha_2$), where the eligibility trace ($\lambda$) determines the degree to which the outcome at the second stage retrospectively transfers to the first stage. In simple terms, the model increases (or decreases) the subjective value of an action at stage 1 proportionally to how positively (or negatively) surprising the outcome was, but discounted by the learning rate that describes the contributions of previous outcomes of that specific action. Conversely, the model-based agent is aware of the structure of the task. Specifically, in trial $t$, with first stage state $s_1$, we define the model-free subjective value of action $a$ as

$$Q_{MF}(a, s_1, t) = Q_{MF}(a, s_1, t) + \alpha_1 \delta_1$$

$$\delta_1 = Q_2(s_2, t) - Q_{MF}(s_1, a, t)$$

and the model-based subjective values of action $a$ as

$$Q_{MB}(s_1, a, t) = T(s_1, a) * Q_2(s_2, t)$$

where $Q_2(s_2, t)$ is the subjective value of the second stage $s_2$ at time $t$ and $T(s_1, a)$ is the transition matrix from first to second stage states. The agents learn the values of the planets, $Q_2$, by

$$Q_2(s_2) = Q_2(s_2) + \alpha_2 \delta_2$$

$$\delta_2 = r - Q_2(s_2)$$

$$Q_{MF}(s_1, a) = Q_{MF}(s_1, a) + \lambda \alpha_1 \delta_2$$

The learning rates at the two stages were either different (M2), or the same (M3).

## Model estimation

We estimated the model parameters for both sessions in one hierarchical (multilevel) Bayesian model. This approach pools information across different levels (drug groups, and participants) and thus leads to more stable individual parameter estimates (*Ahn et al., 2017*), reduces overfitting (*McElreath, 2016*), and enables us to estimate in one model both individual and group level parameters as well as differences between sessions (*Lengersdorff et al., 2020*). Models were implemented in Stan (*Carpenter, 2017*) using R as the interface. Stan uses a Markov Chain Monte Carlo sampling method to describe posterior distributions of model parameters. We ran each candidate model with four independent chains and 3000 iterations (1000 warm-up). Convergence of sampling chains was estimated through the Gelman-Rubin $R$ statistic (*Gelman and Rubin, 1992*), whereby we considered $R$ values smaller than 1.01 as acceptable.

For all subject level parameters (e.g., $\omega$) we drew both the baseline ($\omega_0$) and the session difference ($\Delta\omega$), from a multivariate Gaussian prior. Specifically, for model M1:

$$\begin{pmatrix} \omega_0' \\ \gamma_0' \\ \eta_0' \\ \Delta\omega' \\ \Delta\gamma' \\ \Delta\eta' \end{pmatrix} \sim MVNormal \begin{pmatrix} \mu_{\omega_0'} \\ \mu_{\gamma_0'} \\ \mu_{\eta_0'} \\ \mu_{\Delta\omega'} \\ \mu_{\Delta\gamma'} \\ \mu_{\Delta\eta'} \end{pmatrix}, \ S$$

where $S$ is the covariance matrix, which was factored into a diagonal matrix with standard deviations and the correlation matrix $R$ (*Bürkner, 2017*; *McElreath, 2016*). The prime denotes the parameters in estimation space. The parameters that were fully constrained (i.e., $\omega$, $\alpha_1$, $\alpha_2$, $\lambda$, $\gamma$) were estimated in the inverse probit space and the parameters that only had a lower bound (i.e., $\eta$) were estimated in log space. The hyper-priors for all group-level means were weakly informative, $\mu_{\omega'} \sim N(0,1)$, the prior for group-level standard deviations were $\sigma_{\omega'} \sim HalfNormal(0,1)$, and the prior for the correlation matrix was $R \sim LKJcorr(2)$. The parameters in the second session were therefore defined by their estimated baseline and difference between the sessions. For instance, the model-based weight $\omega$ in the second session for participant $i$ was defined as:

$$\omega(i) = \phi\left(\omega_0'(i) + \Delta\omega'(i)\right)$$

where $\phi$ is the cumulative distribution function of the standard normal distribution (inverse of probit). Effect sizes are calculated by normalizing the relevant regression coefficients by the pooled standard deviation (square root of the sum of all relevant variance components *Nalborczyk et al., 2019*; *Hedges, 2007*). In models where there are no random effects, this reduces to a Cohen's d. For example, the effect sizes for drug effects on $\omega$ were calculated by dividing the estimated difference between group means by the square root of the sum of the variance of both the baseline ($\sigma^2_{\omega_0'}$) and the session difference ($\sigma^2_{\Delta\omega'}$). We used the same procedure in all computational models.

## Model comparison and validation

We used the trial-based Leave-One-Out Information Criterion (LOOIC) to compare the three models using the loo package in R (*Vehtari et al., 2017*). The LOOIC estimates out-of-sample predictive accuracy of each trial and is more informative than simpler point-estimate information criterions used commonly (such as the Akaike information criterion). Lower LOOIC scores indicate better prediction accuracy out of sample. Additionally, we compared models with bootstrapped pseudo Bayesian Model Average relative weights of the models that reflect the posterior probability of each model given the data (*Yao et al., 2018*). To validate the novel model M1, we first used the posterior means of the estimated parameters to simulate behaviour in both sessions. To see how well we can retrieve the model parameters we reran the parameter estimation on the simulated behaviour with the same model (*Figure 4—figure supplement 1 and d*). We also ran the same analysis of staying behaviour on this synthetic dataset and reproduced the behavioural plots from *Figure 3* (compare to *Figure 4—figure*

*supplement 1 and e*). We used the same logistic hierarchical Bayesian model to statistically evaluate the crucial aspects of the behavioural analysis (*Figure 4—figure supplement 1 and e*, compare to *Figure 3c*). To get the posterior predictive accuracy of the model we predicted the choice on each trial for each participant for 8000 samples drawn from the posterior distribution and then calculated the average accuracy for each participant.

## Estimation of the effects of the pharmacological treatment

To statistically evaluate the effect of our treatment on all the parameters in the model, we included two regression terms when defining the group-level means of the difference between sessions:

$$
\begin{pmatrix} \omega'_0 \\ \gamma'_0 \\ \eta'_0 \\ \Delta\omega' \\ \Delta\gamma' \\ \Delta\eta' \end{pmatrix} \sim MVNormal \begin{pmatrix} \mu_{\omega'_0} \\ \mu_{\gamma'_0} \\ \mu_{\eta'_0} \\ \mu_{\Delta\omega'} + \beta^{\omega}_{Ami} * X_{Ami} + \beta^{\omega}_{Nal} * X_{Nal} \\ \mu_{\Delta\gamma'} + \beta^{\gamma}_{Ami} * X_{Ami} + \beta^{\gamma}_{Nal} * X_{Nal} \\ \mu_{\Delta\eta'} + \beta^{\eta}_{Ami} * X_{Ami} + \beta^{\eta}_{Nal} * X_{Nal} \end{pmatrix}, \ S
$$

where $\beta^*_{Ami}$ and $\beta^*_{Nal}$ are coefficients drawn from prior distribution $N(0, 1.5)$ and $X_{Ami}$ and $X_{Nal}$ are dummy variables for the two drugs. To see which of the parameters are affected by the drug treatment, we included group-level effects on all parameters. This model was extended in the second step to include a group-level categorical variable for high/low serum levels of amisulpride (coded as 1 for high serum level and 0 otherwise). In the third step, the model also included the subject-level stickiness parameters and all corresponding group level effects.

To estimate the effects of mood, working memory performance, sex, weight and age, we ran another model predicting estimated differences in $\omega$ and $\eta$ across sessions from those variables. Similarly, in another model, the session differences were predicted by drug variables and their interactions with the four genotype variables. Full model definitions and outputs are provided in supplementary material.

## Reading span task

We used an automated version of the Reading Span Task, where in each block participants saw 2–6 words serially presented that they had to recall by the end of the trial in any order. The words were interlaced with sentences that participants were instructed to judge as either making sense or not (*Klaus and Schriefers, 2016*). Participants played 15 blocks. Correctly recalled items were calculated as a proportion within the block and then averaged across blocks. The effect of the drug treatment was calculated with a Bayesian linear model where the mean score was predicted by drug treatment.

## Acknowledgements

The study was supported by the Vienna Science and Technology Fund (WWTF) with a grant (CS15-003) awarded to Giorgia Silani and Christoph Eisenegger and a grant (VRG13-007) awarded to Christoph Eisenegger and Claus Lamm. We thank Prof. Boris Quednow for his advice on the study design. This work would not be possible were it not for the students who have carried out the data collection: Mani Erfanian Abdoust, Anne Franziska Braun, Raimund Bühler, Lena Drost, Manuel Czornik, Lisa Hollerith, Berit Hansen, Luise Huybrechts, Merit Pruin, Vera Ritter, Frederic Schwetz, Conrad Seewald, Carolin Waleew, Luca Wiltgen, Stephan Zillmer. We thank Catherine Hartley for sending us her version of the task implemented in matlab, and for allowing us to use the stimuli. We are grateful to Wouter Kool for making his version of the task implemented in JsPsych freely available on github. We also thank Lei Zhang and Lukas Lengersdorff for helpful discussions on the statistical procedures used in the manuscript. The manuscript has significantly improved from the constructive and insightful remarks of the three reviewers.

## Additional information

### Funding

| Funder | Grant reference number | Author |
|---|---|---|
| Vienna Science and Technology Fund | CS15- 003 | Giorgia Silani |
| Vienna Science and Technology Fund | VRG13-007 | Christoph Eisenegger |

The funders had no role in study design, data collection and interpretation, or the decision to submit the work for publication.

### Author contributions

Nace Mikus, Conceptualization, Resources, Data curation, Software, Formal analysis, Validation, Investigation, Visualization, Methodology, Writing - original draft, Writing – review and editing; Sebastian Korb, Conceptualization, Data curation, Supervision, Investigation, Methodology, Project administration, Writing – review and editing; Claudia Massaccesi, Data curation, Supervision, Investigation, Project administration, Writing – review and editing; Christian Gausterer, Conceptualization, Resources, Methodology, Writing – review and editing; Irene Graf, Data curation, Supervision, Investigation, Project administration; Matthäus Willeit, Conceptualization, Resources, Supervision, Investigation, Project administration, Writing – review and editing; Christoph Eisenegger, Conceptualization, Funding acquisition; Claus Lamm, Conceptualization, Supervision, Writing – review and editing; Giorgia Silani, Conceptualization, Supervision, Funding acquisition, Project administration, Writing – review and editing; Christoph Mathys, Resources, Software, Supervision, Methodology, Writing – review and editing

### Author ORCIDs

Nace Mikus http://orcid.org/0000-0002-3445-9464
Sebastian Korb http://orcid.org/0000-0002-3517-3783
Claudia Massaccesi http://orcid.org/0000-0003-0519-6324
Christian Gausterer http://orcid.org/0000-0002-4875-5459
Claus Lamm http://orcid.org/0000-0002-5422-0653
Giorgia Silani http://orcid.org/0000-0002-4284-3618
Christoph Mathys http://orcid.org/0000-0003-4079-5453

### Ethics

Experiments were conducted at the Medical University of Vienna and the study was approved by the Ethical Committee of the Medical University of Vienna (EK N. 1393/2017). Informed consent was provided by all research participants.

### Decision letter and Author response

Decision letter https://doi.org/10.7554/eLife.79661.sa1
Author response https://doi.org/10.7554/eLife.79661.sa2

## Additional files

### Supplementary files

• Supplementary file 1. Supplementary Data and Model Summaries. (a) Estimates and CIs of fixed effects of the Bayesian logistic regression predicting staying behaviour. Q2.5 and Q97.5 are the 2.5% and 97.5% quantiles of the posterior parameter distribution. For details on how the posterior distributions were calculated refer to the code online (Ami = Amisulpride, Nal = Naltrexone). (b) Results of a Bayesian logistic linear model predicting percentage correct from drug groups.(c) Mood in mean and standard deviation at time of pill intake (T1) and 3 hr later (T2).(d) Drug effects on differences in positive PANAS scales (centralized) between sessions. Drug variables coded as follows: nal is as dummy variable for naltrexone (1 for naltrexone, 0 otherwise), ami is a dummy variable for amisulpride, and serum_ami is a dummy variable for serum (1 only in the high serum group, and 0 otherwise). (e) Drug effects on differences in negative PANAS scales (centralized) between sessions. Drug variables coded as before. (f) Drug effects on session differences in $\omega$,

including mood at baseline, difference in mood from baseline and working memory performance as covariates, as well as sex, age and weight as moderators of effects. Drug variables coded as before, all other dependent variables scaled and centralized. (g) Drug effects on session differences in $\eta$, including mood at baseline, difference in mood from baseline and working memory performance as covariates, as well as sex, age and weight as moderators of effects. Drug variables coded as before, all other dependent variables scaled and centralized. (h) Drug effects on session differences in $\omega$, from genetic variables. Drug variables coded as before, all other dependent variables scaled and centralized. (i) Drug effects on session differences in $\omega$, from genetic variables. Drug variables coded as before, all other dependent variables scaled and centralized. (j) Description of participants in terms of body mass index (BMI), age and sex with mean (m) and standard deviation (sd). (k) Distribution of genotypes. (l) Number of participants per drug group used in analysis.

- MDAR checklist

### Data availability

The data and the analysis scripts are available at https://github.com/nacemikus/mbmf-da-op.git, (copy archived at swh:1:rev:4822b12aa33d8e5eb60d8ad5af2a0d3392e00e20).

The following dataset was generated:

| Author(s) | Year | Dataset title | Dataset URL | Database and Identifier |
|---|---|---|---|---|
| Mikus N, Korb S, Massaccesi C, Gausterer C, Graf I, Willeit M, Eisenegger C, Lamm C, Silani G, Mathys C | 2022 | Making data and code from the article "Effects of dopamine D2 and opioid receptor antagonism on the trade-off between model-based and model-free behaviour in healthy volunteers" available online. | https://doi.org/10.5281/zenodo.7221355 | Zenodo, 10.5281/zenodo.7221355 |

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
