## [Editor Report]

This study provides novel evidence that a dopamine D2/D3 receptor antagonist enhances model-based control of behavior, whereas blocking opioid receptors has no effect. These conclusions are based on compelling behavioral and computational modeling data. The paper makes an important contribution to our understanding of how dopamine shifts the balance between two subsystems regulating behavior and may improve the understanding of motivational dysfunctions in mental disorders like addiction.

---

## [Decision Letter]

**Decision letter after peer review:**

Thank you for submitting your article "Effects of dopamine D2 and opioid receptor antagonism on the trade-off between model-based and model-free behaviour in healthy volunteers" for consideration by *eLife*. Your article has been reviewed by 3 peer reviewers, one of whom is a member of our Board of Reviewing Editors, and the evaluation has been overseen by a Reviewing Editor and Michael Taffe as the Senior Editor. The reviewers have opted to remain anonymous.

Essential revisions:

The reviewers agree that your paper is solid, timely, and interesting but that a few issues have to be addressed.

Please address the comments in the individual reviews, with a particular emphasis on the following points:

1) Provide a better motivation for studying the effects of these two drugs and specific predictions for their effects.

2) Revise the statistical analysis to include additional random effects.

3) Discuss the issue that baseline and drug sessions are not counter-balanced as limitations.

4) Include drug serum levels in the analysis.

5) Elaborate on the detrimental effects of amisulpride on stay-choices when the first stage stays the same.

*Reviewer #1 (Recommendations for the authors):*

Please include serum levels for amisulpride and naltrexone should be included as covariates in the analysis.

It's not entirely clear to me how the detrimental effect of amisulpride on high-point repeat choices when the first stage stays the same (which I believe is captured by the inverse temperature) can be reconciled with the idea that blocking D2 receptors reduces flexibility and enhances the stability of prefrontal representations.

If I understand the models correctly, the only difference between the three models is the number of free learning rates. The learning rate in M1 is set to 1, and M2 and M3 allow two or one free learning rate. If this is correct, it would be easier to just describe it like that rather than presenting them as fundamentally different models.

*Reviewer #2 (Recommendations for the authors):*

1. One motivation for the study is that dopamine agonists yielded inconsistent results, but also the effects of D2 antagonists like amisulpride are not straightforward to interpret. This is because the administered dose of 400 mg can lead to either presynaptic effects (which increases dopaminergic activity) or postsynaptic effects (reducing dopaminergic activity, as assumed by the authors). Participants with a lower effective dose may show stronger presynaptic than postsynaptic effects, and vice versa for individuals with a higher effective dose. To control for the effective dose, the authors might use the measured plasma concentrations. If the current interpretation is correct, the impact of amisulpride on the weighing parameter should increase with higher plasma concentrations, suggesting that the observed mean effect would be driven by postsynaptic rather than presynaptic mechanisms.

2. According to supplementary Table 1, only the main effects of "session", "prev_state_diff", and "prev_points" were modelled as random slopes, whereas as fixed effects also all interaction terms were modelled. As it is generally recommended to maximize the random effects matrix in order to reduce the risk of false positives, I ask the authors to model also the interaction terms between these variables as random slopes. Does this change the observed amisulpride effects?

3. Why did the authors control for BMI (supplementary Table 5)? If the intention was to control for the effective individual dose, body weight would be more relevant than BMI.

4. The inverse temperature parameter in the computational model is described as an indicator of both exploratory behavior and decision noise. Is exploratory behavior the same as decision noise in the current task, or does one parameter indeed measure two dissociable constructs?

*Reviewer #3 (Recommendations for the authors):*

Here are two small comments about the analyses. First, why didn't the authors include stickiness parameters in the model? This is very popular in our field because it allows us to capture variance in choice induced by preferences that are not related to maximizing reward. Second, the behavioral analysis that tests the effect of a previous reward on choice is not quite ideal, because it doesn't take into account that the same scalar reward (e.g., +1 point) can elicit a negative prediction error (if you expected +5) or a positive prediction error (if you expected -3). This issue can be remedied by computing prediction errors for the second-stage reward and then using their sign as a predictor of staying behavior.

[Editors' note: further revisions were suggested prior to acceptance, as described below.]

Thank you for resubmitting your work entitled "Effects of dopamine D2 and opioid receptor antagonism on the trade-off between model-based and model-free behaviour in healthy volunteers" for further consideration by *eLife*. Your revised article has been evaluated by Michael Taffe (Senior Editor) and a Reviewing Editor.

The manuscript has been improved but there are some remaining issues that need to be addressed, as outlined below:

As suggested by reviewer #2, please repeat the analyses using continuous rather than categorical plasma levels (or justify the use of a categorical variable), and also include plasma levels for the model-free behavioral analyses.

*Reviewer #1 (Recommendations for the authors):*

The authors have done a very nice job addressing my initial comments. I don't have any additional points.

*Reviewer #2 (Recommendations for the authors):*

The authors made a good job of revising the manuscript and successfully addressed almost all of my previous concerns. However, I still have some questions regarding the inclusion of plasma concentrations in the statistical analyses: First, I wondered why plasma concentrations were dichotomized into low versus high (and was this done based on the mean or the median)? It seems more straightforward to enter plasma concentrations as continuous predictors to the models, as this takes the full variation in plasma concentrations between individuals into account. Second, it seems that plasma concentrations were added only to the model-based computational analyses, but in order to be consistent the authors should do the same with their model-free behavioral analyses.

---

## [Author Response]

Reviewer #1 (Recommendations for the authors):Please include serum levels for amisulpride and naltrexone should be included as covariates in the analysis.

We thank the reviewer for the excellent suggestion to also include serum levels as covariates. Doing so had a profound impact on the interpretation of the results of our study. By including the serum variable, it became clear that the increased exploration is predominantly observed in participants with high amisulpride serum levels in the blood, whereas the increase in the model-based weight did not scale with serum levels (see Figure 5). We note also that the blood samples of the naltrexone group were not analysed, based on the finding that 50 mg provides abundant (~90%) and consistent occupancy of opioid receptors (Trøstheim, Eikemo, Haaker, Frost, and Leknes, 2022).

The following text is now included in the results (p.6 §3,4):

“We next looked at whether the effects of amisulpride were different based on the effective dose. We reran the parameter estimation including a variable for amisulpride blood serum levels (Figure 5). We found that amisulpride increased ω in both the low serum group (b = 0.919, 95% CI [0.216, 1.722], P(b<0) = 0.006, d = 1.013, 95% CI [0.238, 1.897]) and the high serum group (b = 0.872, 95% CI [0.008, 1.853], P(b<0) = 0.024, d = 0.961, 95% CI [0.009, 2.041]), with no difference between the effects (P(b>0) = 0.458). However, when looking at the effects of amisulpride on η we found that it was not reduced in the low serum group (b = -0.105, 95% CI [-0.523, 0.348], P(b>0) = 0.323, d = -0.096, 95% CI [-0.477, 0.317]), but was reduced in the high serum group (b = -0.492, 95% CI [-0.96, -0.033], P(b>0) = 0.018, d = -0.45, 95% CI [-0.877, -0.03]), with a (non-significant) moderate effect size difference between the effects (b = -0.393, 95% CI [-0.918, 0.118], P(b>0) = 0.066, d = -0.359, 95% CI [-0.838, 0.108]).

Lower values of η imply more choice behaviour that the model cannot explain based on estimated values of the two spaceships. To see whether some of the increased variance can be explained by the tendency of participants to repeat actions or choices regardless of outcomes, we extended the M1 serum model to include a response stickiness (ρ) and stimulus stickiness (π) parameter. We found some evidence that amisulpride in participants with high serum levels increased switching between responses, as indicated by the reduced ρ parameter (b = -0.466, 95% CI [-0.962, 0.043], P(b>0) = 0.039). This effect was not present in the low serum group (b = -0.158, 95% CI [-0.618, 0.311], P(b>0) = 0.251), with a non-significant difference between the effects (b = -0.294, 95% CI [-0.723, 0.13], P(b>0) = 0.082). The inclusion of the two stickiness parameters, did not markedly change the posterior distributions of effects of amisulpride on η, but did reduce the effect on ω in the high serum group (b = 0.571, 95% CI [-0.361, 1.527], P(b<0) = 0.111, see Figure 4 —figure supplement 2 for other relevant posterior distributions). We note however, that the model including stickiness parameters performed worse than the model without the stickiness parameters, with a lower out-of-sample trial-wise predictive performance (Δelpd(sd) = -413 (31.5)).”

It's not entirely clear to me how the detrimental effect of amisulpride on high-point repeat choices when the first stage stays the same (which I believe is captured by the inverse temperature) can be reconciled with the idea that blocking D2 receptors reduces flexibility and enhances the stability of prefrontal representations.

We agree with the reviewer that this issue was not sufficiently discussed in the manuscript. As the reviewer points out – we now show that the subject-level random slope of “high-point repeat choices when the first stage stays the same” is indeed reflected in the inverse temperature parameter (p.6, §4). With the addition of serum data, we can speculate that the exploration could be mainly due to occupation of the post-synaptic D2 receptors, which are known to be involved in the exploration-exploitation trade-off and choice stochasticity. We also show that the random slope for different first stage stays is more related to the model-based/model-free weight. As mentioned above, this parameter does not scale with the serum levels suggesting the effect is not post-synaptic and is present already with small doses. This might imply that the increase in model-based weight is driven by presynaptic effects. However, there is also a line of research (the dual state theory of prefrontal dopamine function proposed by Durstewitz and Seamans, 2008) showing the role of prefrontal D2 in destabilisation of prefrontal representations. We therefore also present the argumentation that D2 receptor antagonists might increase stability of prefrontal representations which in turn helps explain the ability of subjects under amisulpride to maintain the mapping online. As the reviewer alluded to, this take on the effects is hard to reconcile with increased exploration and also with flexible adaptations of choice selection and fast action values updating that is required in this task. We elaborate on this in the discussion (p.8 §2):

“According to the dual-state theory of prefrontal dopamine function (Durstewitz and Seamans, 2008), a state dominated by D1 receptor activity is characterized by a high-energy barrier and supports stabilization of prefrontal representations. Conversely, a D2 receptor dominated state has a low-energy barrier and facilitates flexible switching between representational states. In line with this, D2 antagonists impair behaviour in tasks that require constant attentional set shifting (Mehta, Manes, Magnolfi, Sahakian, and Robbins, 2004), but might improve performance in tasks where prefrontal goal representations are required (Kahnt, Weber, Haker, Robbins, and Tobler, 2015). For example, D2 antagonism improves the ability of participants in certain cognitive tasks that require manipulation of task-relevant information in the working memory (Dodds et al., 2009; Frank and O’Reilly, 2006), but does not improve working memory capacity, or memory retrieval (Dodds et al., 2009; Mehta, Sahakian, McKenna, and Robbins, 1999; Naef et al., 2017). The effects of D2 antagonism on model-based/model-free behaviour in our study can be interpreted within this framework to result from increased ability to maintain prefrontal representation of the mapping between the spaceships and the planets online. However, this is difficult to reconcile with the fact that model-based behaviour in dynamic learning paradigms, such as the one used here, also requires flexible updating of action values.”

If I understand the models correctly, the only difference between the three models is the number of free learning rates. The learning rate in M1 is set to 1, and M2 and M3 allow two or one free learning rate. If this is correct, it would be easier to just describe it like that rather than presenting them as fundamentally different models.

We agree that the relationship between the three models should be explained better. The M1 model is indeed similar to the M2 and M3 models with fixed learning rates but also included the devaluation parameter on the not chosen option and omits the eligibility trace present in both M2 and M3 models. This has now been elaborated in the computational modelling part of the Results section.

The corresponding text (p. 5 §3) reads as follows:

“We compared the above-described model to two Dual-systems Reinforcement Learning (RL) models (Figure S1a). In these two models, the degree to which an outcome in each trial affects the subjective values of actions at each stage is determined by a learning rate parameter. The model-free agent thus learns the subjective action values in each stage from experience, by increasing the values of actions and states that lead to outcomes that were better than expected and decreasing the values when the outcome was worse than expected. We allow the learning rate at both stages to either be different (model M2) or the same (model M3). Note, that the model M1 is a version of these RL models, where the learning rates are set to 1 and a devaluation parameter is included on the non-chosen option. This is motivated by the observation that the rewards change according to a Gaussian random walk (and are not probabilistic), and therefore the last encountered outcome is the best guess the agent can make. When comparing the performance of the three models we found that the model M1 has better out of sample predictive accuracy compared to the other two models (Figure S1b). We verified the winning model with parameter recovery (Figure S1c) and posterior predictive checks (Figure S1d-f).”

Reviewer #2 (Recommendations for the authors):1. One motivation for the study is that dopamine agonists yielded inconsistent results, but also the effects of D2 antagonists like amisulpride are not straightforward to interpret. This is because the administered dose of 400 mg can lead to either presynaptic effects (which increases dopaminergic activity) or postsynaptic effects (reducing dopaminergic activity, as assumed by the authors). Participants with a lower effective dose may show stronger presynaptic than postsynaptic effects, and vice versa for individuals with a higher effective dose. To control for the effective dose, the authors might use the measured plasma concentrations. If the current interpretation is correct, the impact of amisulpride on the weighing parameter should increase with higher plasma concentrations, suggesting that the observed mean effect would be driven by postsynaptic rather than presynaptic mechanisms.

We thank reviewer 2 for this constructive criticism and have now included serum levels in the analysis. This turned out to be of great value to pinpoint the effects of amisulpride. The newly provided Figure 5 (see below) shows that increased model-based weight did not scale with serum levels in the blood and was present in participants with either low or high serum levels. Further, we found that the higher exploration is predominantly observed in participants with high amisulpride serum levels in the blood and was not present in the low-serum group. Given that amisulpride might act on post-synaptic receptors with a higher effective dose this would be in-line with the well-established model of the role of postsynaptic D2 receptors corticostriatal loops in action selection and exploration. Our finding that we find an increase in the model-based weight already at low effective doses, might support the notion that the effect might be related to amisulpride increasing availability of dopamine in the synaptic cleft through blockade of the D2 autoreceptors.

These results are now described in the results (p.8-9) and in Figure 5:

“We next looked at whether the effects of amisulpride were different based on the effective dose. We reran the parameter estimation including a variable for amisulpride blood serum levels (Figure 5). We found that amisulpride increased ω in both the low serum group (b = 0.919, 95% CI [0.216, 1.722], P(b<0) = 0.006, d = 1.013, 95% CI [0.238, 1.897]) and the high serum group (b = 0.872, 95% CI [0.008, 1.853], P(b<0) = 0.024, d = 0.961, 95% CI [0.009, 2.041]), with no difference between the effects (P(b>0) = 0.458). However, when looking at the effects of amisulpride on η we found that it was not reduced in the low serum group (b = -0.105, 95% CI [-0.523, 0.348], P(b>0) = 0.323, d = -0.096, 95% CI [-0.477, 0.317]), but was reduced in the high serum group (b = -0.492, 95% CI [-0.96, -0.033], P(b>0) = 0.018, d = -0.45, 95% CI [-0.877, -0.03]), with a (non-significant) moderate effect size difference between the effects (b = -0.393, 95% CI [-0.918, 0.118], P(b>0) = 0.066, d = -0.359, 95% CI [-0.838, 0.108]).

Lower values of η imply more choice behaviour that the model cannot explain based on estimated values of the two spaceships. To see whether some of the increased variance can be explained by the tendency of participants to repeat actions or choices regardless of outcomes, we extended the M1 serum model to include a response stickiness (ρ) and stimulus stickiness (π) parameter. We found some evidence that amisulpride in participants with high serum levels increased switching between responses, as indicated by the reduced ρ parameter (b = -0.466, 95% CI [-0.962, 0.043], P(b>0) = 0.039). This effect was not present in the low serum group (b = -0.158, 95% CI [-0.618, 0.311], P(b>0) = 0.251), with a non-significant difference between the effects (b = -0.294, 95% CI [-0.723, 0.13], P(b>0) = 0.082). The inclusion of the two stickiness parameters, did not markedly change the posterior distributions of effects of amisulpride on η, but did reduce the effect on ω in the high serum group (b = 0.571, 95% CI [-0.361, 1.527], P(b<0) = 0.111, see Figure 4 —figure supplement 2 for other relevant posterior distributions). We note however, that the model including stickiness parameters performed worse than the model without the stickiness parameters, with a lower out-of-sample trial-wise predictive performance (Δelpd(sd)= -413 (31.5)).”

See also the discussion of these results (p.9 §1 and §2) that includes the following text:

“An important aspect of drugs that target D2 receptors is that they are known to mainly act on presynaptic D2 receptors at low doses, while binding to postsynaptic receptors prevailingly accounts for their effects at higher doses (Schoemaker et al., 1997). Presynaptic D2 receptors are believed to play an autoregulatory control of dopamine signalling through inhibition of synthesis and dopamine release (Ford, 2014). In our data, we find that the increase in model-based control did not scale with higher effective dose and was present in participants with either high or low serum levels in the blood. The effect in the high serum group was slightly reduced (and not robustly above zero, p = 0.111) in the computational model that included stickiness parameters, although we note that including stickiness parameters led to a poorer model fit. Taken together, this provides some evidence for the notion that the effects of amisulpride on model-based behaviour are not driven by postsynaptic effects but might rather be due to amisulpride increasing dopamine levels through presynaptic D2 receptor blockade and thereby boosting the willingness to apply cognitive effort.

Interestingly, amisulpride also increased choice stochasticity parametrised by the softmax inverse temperature parameter. In a paradigm with two choice options, it cannot be definitively determined whether this indicates higher decision-noise or increased exploration of alternative choices. We can however speculate that increased decision noise would lead to overall detrimental effects on learning in both trial types with same and different consecutive first stage states, which we do not observe in our data. The effect on the choice stochasticity parameter was only present in participants with a higher effective dose (Rosenzweig et al., 2002), suggesting that the effect was more likely to be post-synaptic. Similarly, in the same effective dose group, we found some evidence that amisulpride reduces response stickiness indicating increased switching between actions. This is well in line with a prominent model of the cortico-striatal circuitry implicating post-synaptic D2 receptors in exploration/exploitation (Frank, 2005) and supported by empirical data.”

2. According to supplementary Table 1, only the main effects of "session", "prev_state_diff", and "prev_points" were modelled as random slopes, whereas as fixed effects also all interaction terms were modelled. As it is generally recommended to maximize the random effects matrix in order to reduce the risk of false positives, I ask the authors to model also the interaction terms between these variables as random slopes. Does this change the observed amisulpride effects?

We thank the reviewer for this suggestion. We now use the model with the interactions in the manuscript, and the main findings have not changed. However, we note that the effect of amisulpride on trials where the first stage state was different (βlogodds=0.046 (95% CI [-0.007, 0.098], P(βlogodds<0) = 0.048) has slightly changed, and is now lower and has more uncertainty around it – i.e. a larger CI (βlogodds=0.036 (95% CI -0.091, 0.158], P(βlogodds<0) = 0.285). The interaction effect is still robust as is the effect of amisulpride on same first state trials. When comparing the random slopes from this model with computational parameters we see that the exploration parameter mostly explains choices in the same first state trials, while model-based weight explains the difference between the same and different first state trials (p.6 §1). We therefore interpret our findings as amisulpride having a general (likely a post-synaptic) effect on choice performance by increasing exploration, while still relatively boosting model-based relative to model-free behaviour.

3. Why did the authors control for BMI (supplementary Table 5)? If the intention was to control for the effective individual dose, body weight would be more relevant than BMI.

BMI was used in line with previous studies, but we agree with the reviewer that normalizing the weight by height makes less sense when considering effective doses. We now use weight in our analysis. Nothing changed though in our main conclusions, as neither weight nor BMI affects any of the inferences about the main outcome variables.

4. The inverse temperature parameter in the computational model is described as an indicator of both exploratory behavior and decision noise. Is exploratory behavior the same as decision noise in the current task, or does one parameter indeed measure two dissociable constructs?

Unfortunately, in this task the behavioural patterns arising from decision-noise and exploratory behaviour are not separable. We can only speculate that if amisulpride would increase decision noise, we would find a more generally detrimental effect on behaviour. It is therefore more likely that amisulpride increased choice switching due to higher drive for exploring other options.

This is now mentioned in the discussion (p.9 §2):

“Interestingly, amisulpride also increased choice stochasticity parametrised by the softmax inverse temperature parameter. In a paradigm with two choice options, it cannot be definitively determined whether this indicates higher decision-noise or increased exploration of alternative choices. We can however speculate that increased decision noise would lead to overall detrimental effects on learning in both trial types with same and different consecutive first stage states, which we do not observe in our data. The effect on the choice stochasticity parameter was only present in participants with a higher effective dose (Rosenzweig et al., 2002), suggesting that the effect was more likely to be post-synaptic. Similarly, in the same effective dose group, we found some evidence that amisulpride reduces response stickiness indicating increased switching between actions. This is well in line with a prominent model of the cortico-striatal circuitry implicating post-synaptic D2 receptors in exploration/exploitation (Frank, 2005) and supported by empirical data. In animal studies, activation of D2 receptors was shown to lead to choice perseverance and more deterministic behaviour, whereas D2 receptor inhibition increases the probability of performing competing actions and increases randomness in action selection (Sridharan, Prashanth, and Chakravarthy, 2006). In humans, a recent neurochemical imaging study showed that D2 receptor availability in the striatum correlated with choice uncertainty parameters across both reinforcement learning and active inference computational modelling frameworks (Adams et al., 2020). Increased choice uncertainty was also observed in a social and non-social learning tasks in a study using 800 mg of sulpiride, a dose that is known to exert post-synaptic effects (Eisenegger et al., 2014; Mikus et al., 2022).”

Reviewer #3 (Recommendations for the authors):Here are two small comments about the analyses. First, why didn't the authors include stickiness parameters in the model? This is very popular in our field because it allows us to capture variance in choice induced by preferences that are not related to maximizing reward. Second, the behavioral analysis that tests the effect of a previous reward on choice is not quite ideal, because it doesn't take into account that the same scalar reward (e.g., +1 point) can elicit a negative prediction error (if you expected +5) or a positive prediction error (if you expected -3). This issue can be remedied by computing prediction errors for the second-stage reward and then using their sign as a predictor of staying behavior.

We thank the reviewer for suggesting including the stickiness parameters. Although the results did not change much we do find some support for the fact that amisulpride increased switching between responses in individuals with high serum levels in the blood. This lends further support that higher doses of D2 antipsychotics might affect action selection policies. Regarding the reviewer’s second point, we agree that looking at the effects of previous points on staying behaviour is not ideal and might not account for cases where same reward sizes encourage or discourage choice repetition. This is exactly what motivates the computational modelling approach on top of using simple logistic regression to predict staying behaviour. We elaborate on this in the Results section on p.5:

“We note also that this analysis approach ignores one important aspect of the task, namely that subjects were comparing the relative points between two independently changing planets, thereby the points received in each trial should be seen in relation to the points that the participants expected to get were they to choose the other spaceship. To address these issues, we employed computational modelling.”

[Editors' note: further revisions were suggested prior to acceptance, as described below.]

The manuscript has been improved but there are some remaining issues that need to be addressed, as outlined below:As suggested by reviewer #2, please repeat the analyses using continuous rather than categorical plasma levels (or justify the use of a categorical variable), and also include plasma levels for the model-free behavioral analyses.Reviewer #2 (Recommendations for the authors):The authors made a good job of revising the manuscript and successfully addressed almost all of my previous concerns. However, I still have some questions regarding the inclusion of plasma concentrations in the statistical analyses: First, I wondered why plasma concentrations were dichotomized into low versus high (and was this done based on the mean or the median)? It seems more straightforward to enter plasma concentrations as continuous predictors to the models, as this takes the full variation in plasma concentrations between individuals into account. Second, it seems that plasma concentrations were added only to the model-based computational analyses, but in order to be consistent the authors should do the same with their model-free behavioral analyses.

We thank the reviewer for pointing out the issue with plasma concentrations. We briefly explained in the Methods that the serum variable was dichotomized due to the skew in the distribution. Admittedly this was perhaps not spelled out clearly enough. We, thus, now mention this in the main text and describe the rationale in detail in the Procedure section of the Methods.

In results (p.6) we now say:

“We next looked at whether the effects of amisulpride were different based on the effective dose. We reran the parameter estimation including a variable for amisulpride blood serum levels as a group-level covariate in the hierarchical model (Figure 5). We used a categorical variable due to the highly skewed distribution of serum levels (DeCoster et al., 2011, see Methods for details).”

In the methods we elaborate on this on p.10:

“For 18 participants the value of amisulpride was at maximum (>604 μg/l), leading to a highly skewed distribution, with no variance in the upper 50% quantile. Because of this we dichotomized the data into a categorical variable, whereby the high serum group consisted of participants with serum levels above 604 μg/l (n = 18) and the low serum group had levels between 212.6 μg/l and 602.5 μg/l (n = 14).”

We also include the behavioural (“model-free”) analysis with serum levels which we agree adds to the consistency of the manuscript. The following paragraph has been added to the Results section (p.):

”We also reran the behavioural analysis, predicting the likelihood to stay with the previous choice as before, but including the serum variable. In line with the results from the computational model, we found that amisulpride significantly increased the difference between the effects of previous points on staying behaviour in different vs. same first state trials both in the low serum group (βlogodds = 0.235, 95% CI [0.047, 0.43], P(βlogodds<0) = 0.006), as well as in the high serum group, although to a lesser extent (βlogodds = 0.143, 95% CI [-0.03, 0.323], P(βlogodds<0) = 0.054). Conversely, in the high serum group, amisulpride decreased the effect of previous points in the same first state trials (b = -0.172, 95% CI [-0.34, -0.002], P(βlogodds>0) = 0.024), mirroring the effect of the computational model. In the low serum group amisulpride also decreased the effect of previous points on staying behaviour in the first state trials (βlogodds = -0.147, 95% CI [-0.34, 0.036], P(βlogodds>0) = 0.060). Note that, although the 95% CI contained values above 0, this represents a slight inconsistency to the results from the computational model, where the exploration parameter η was not reliably reduced in the low serum group.”